# The mechanism for directional hearing in fish

Johannes Veith[1,2,6], Thomas Chaigne[1,3,6], Ana Svanidze[1], Lena Elisa Dressler[1,4], Maximilian Hoffmann[1,5], Ben Gerhardt[1,2] & Benjamin Judkewitz[1✉]

Locating sound sources such as prey or predators is critical for survival in many vertebrates. Terrestrial vertebrates locate sources by measuring the time delay and intensity difference of sound pressure at each ear[1–5]. Underwater, however, the physics of sound makes interaural cues very small, suggesting that directional hearing in fish should be nearly impossible[6]. Yet, directional hearing has been confirmed behaviourally, although the mechanisms have remained unknown for decades. Several hypotheses have been proposed to explain this remarkable ability, including the possibility that fish evolved an extreme sensitivity to minute interaural differences or that fish might compare sound pressure with particle motion signals[7,8]. However, experimental challenges have long hindered a definitive explanation. Here we empirically test these models in the transparent teleost *Danionella cerebrum*, one of the smallest vertebrates[9,10]. By selectively controlling pressure and particle motion, we dissect the sensory algorithm underlying directional acoustic startles. We find that both cues are indispensable for this behaviour and that their relative phase controls its direction. Using micro-computed tomography and optical vibrometry, we further show that *D. cerebrum* has the sensory structures to implement this mechanism. *D. cerebrum* shares these structures with more than 15% of living vertebrate species, suggesting a widespread mechanism for inferring sound direction.

Whether it is the song of a whale, the rustling of a mouse or the quiet sneaking of a cat, when sound emanates from a source, it travels through the surrounding medium as an oscillation of motion and pressure. The ability to detect its direction turns hearing into a spatial sense that is pivotal for survival.

Terrestrial vertebrates sense direction by sampling pressure at the ears (that is, two positions far apart; Fig. 1a). In the late 1940s, Lloyd Jeffress speculated that a combination of delay lines and coincidence detector neurons could measure the interaural time difference (ITD)[1]. An implementation of the Jeffress model was later found by Carr and Konishi in their seminal work on the barn owl brainstem[2], making it one of the most well-known canonical circuits in vertebrates and a textbook example for the convergence of theory, behaviour, biophysics and physiology[3,4]. In addition to the ITD, the difference in the pressure amplitude across the ears, called the interaural level difference (ILD), also carries information about the sound direction. This quantity is amplified by the head, which reflects airborne sound and casts an acoustic shadow onto the far ear[5].

The vertebrate sense of hearing evolved from a common fish ancestor, and yet these well-established models for directional hearing falter when applied to underwater environments. As sound travels approximately five times faster in water than in air, ITDs are reduced to very small levels (Fig. 1b). As biological tissues have acoustic impedances similar to water, ILDs are substantially smaller, too. This presents a

conundrum: according to prevailing models, fish should not be able to localize sound. Yet, behavioural evidence shows that fishes such as the Atlantic cod[7,8,11–14], the plainfin midshipman[15–19], herring[20–23] and goldfish[24–30] can determine the direction of sound sources.

What cues are available to fish that might enable directional hearing? Fish have two distinct peripheral auditory pathways[31,32]. First, otolithic end organs of the inner ear, in addition to their vestibular function, also act as particle motion sensors for nanometre to micrometre displacements. Owing to the morphology of hair cells, this direct hearing pathway is inherently spatial, but it allows animals to tell only the axis of sound propagation, not the direction—a limitation termed the 180° ambiguity problem of directional hearing[6]. The second, so-called indirect, hearing pathway relies on the swim bladder, which is filled with compressible gas that oscillates in a pressure field. In Otophysi, a large superorder containing about 66% of freshwater fish species[33], a series of bones (Weberian ossicles) transmits this motion to the inner ear[34–39].

In 1975, Arie Schuijf proposed a model[7,8] suggesting that fish could theoretically deduce the direction of sound if they were able to separately sense and compare its motion and pressure components. Alternative hypotheses include the possibility that fish evolved an extreme sensitivity to minuscule ITDs and ILDs or use their mechanosensory lateral line organ for directional inference through an unknown mechanism. Despite almost a century of careful work on this topic

[1]Einstein Center for Neurosciences, Charité – Universitätsmedizin Berlin, Berlin, Germany. [2]Institut für Biologie, Humboldt-Universität zu Berlin, Berlin, Germany. [3]Aix Marseille Univ, CNRS, Centrale Med, Institut Fresnel, Marseille, France. [4]Present address: Museum für Naturkunde Berlin, Berlin, Germany. [5]Present address: Rockefeller University, New York, NY, USA. [6]These authors contributed equally: Johannes Veith, Thomas Chaigne. ✉e-mail: benjamin.judkewitz@charite.de

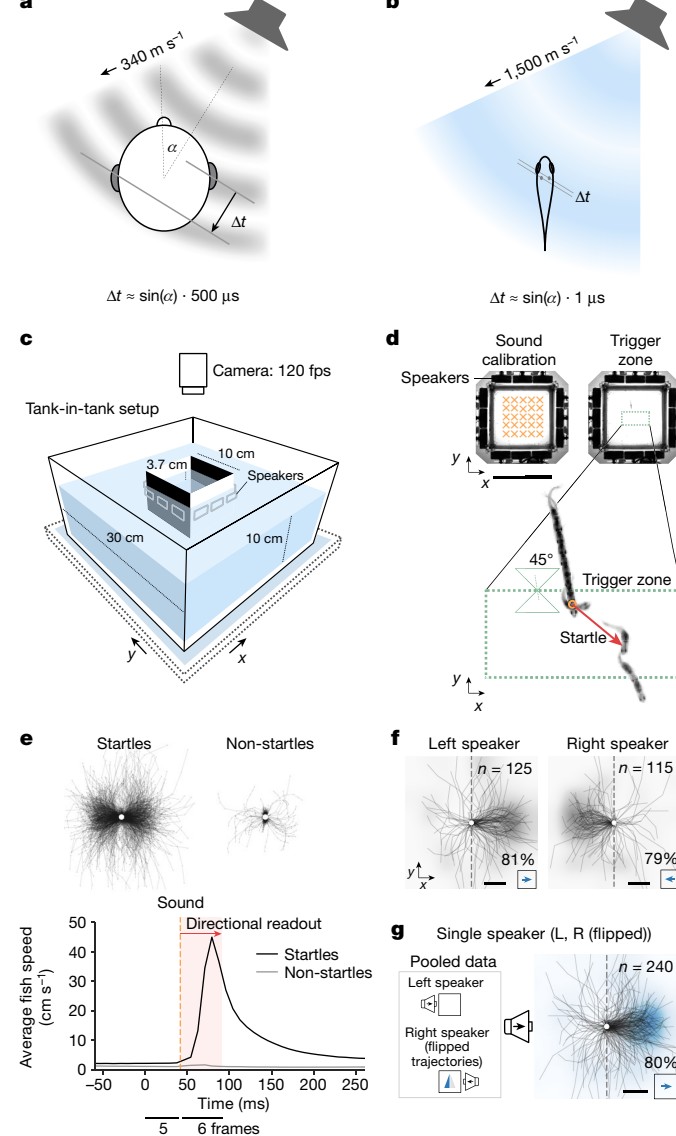

**Fig. 1 | Sounds elicit a directional startle reflex in *Danionella cerebrum*.**
**a**, Schematic of a pressure wave, arriving at the auditory organs with a detectable ITD in humans. **b**, ITDs are heavily diminished underwater (value approximated for *D. cerebrum*). **c**, Behavioural setup (Methods). **d**, Playback paradigm. Before the experiment, sound pressure and particle acceleration are calibrated at multiple points inside the inner tank (top left, orange crosses; see also Extended Data Fig. 1c–e). Playback is triggered if three conditions are met: the fish swims into the trigger zone (top right, dotted green rectangle), the fish is oriented ≤45° to the *y* axis, and ≥5 s have passed since the last playback (Methods). **e**, Startles are detected by a speed threshold after sound playback (Methods; see Extended Data Fig. 4 for details on startle dynamics). Top: centred trajectories after playback for startles (*n* = 1,415) and non-startles (*n* = 2,383) across all fish (*n* = 65). Bottom: average fish speed for startles and non-startles, aligned to sound trigger at *t* = 0. **f**, Centred startle trajectories in two sound configurations show a directional escape away from the left (81% of *n* = 125 startle trials across 58 fish; two-sided binomial test: *P* = 2 × 10⁻¹²) or right (79% of *n* = 115 startle trials across 56 fish; two-sided binomial test: *P* = 2 × 10⁻¹⁰) speaker. **g**, Pooled centred trajectories from **f** with flipped trajectories for the right speaker stimulus summarize the directional escape away from the single speaker (80% of *n* = 240 startle trials across 63 fish; two-sided binomial test: *P* = 1 × 10⁻²¹). **f**,**g**, The heat maps are normalized and smoothed two-dimensional histograms over endpoint positions of the trajectories (grey for single-stimulus data; blue for pooled data). Scale bars, 10 cm (**d**) and 5 mm (**f**-**g**).

(Supplementary Table 1), starting with Karl von Frisch's studies on minnows in 1935[40], the mechanism of directional hearing in fish is debated to this date.

Empirical tests of the directional hearing mechanism have been fraught with several difficulties. First, reliably controlling sound stimuli underwater is challenging owing to echoes and sound reverberations, as well as near-field effects in small tanks[41–45]. Second, although fish are sensitive to particle motion, many studies control only for pressure, so several authors have urged experimenters to control both quantities[41,46,47]. Third, the sense of hearing, mediated by the inner ear, must be distinguished from the lateral line sense, which is sensitive to low-frequency water flow (<200 Hz)[48–51]. Fourth, behavioural paradigms should exclude the possibility of klinotaxis (that is, a sequential gradient sampling strategy) instead of true directional hearing. Finally, the study of sound transduction to the fish's inner ear[36,38,39] is complicated by tissue opacity.

Here we address these challenges and systematically test hypotheses of directional hearing mechanisms using the transparent fish *Danionella cerebrum*, one of the smallest known vertebrates[9,10,52–59]. Its small size makes *D. cerebrum* well suited for high-throughput experiments under controlled laboratory conditions. Moreover, *D. cerebrum* communicate acoustically, underlining the importance of hearing for their behaviour. With an inner ear separation of less than 1 mm, *D. cerebrum* put interaural comparison mechanisms to their ultimate test (Fig. 1b).

We find that *D. cerebrum* perform directional startles away from a sound source and that this ability is independent of the lateral line. We then present an extensive set of controlled particle motion and pressure stimuli that lead to differential predictions for directional responses, depending on seven alternative hypotheses for the mechanism of directional hearing. Finally, we carry out laser-scanning vibrometry of auditory structures across the transparent body to determine the physical cues available to *D. cerebrum*. Together, the findings of our experiments lead us to reject all but one of the proposed hypotheses for directional hearing, and they provide strong support for Schuijf's model that fish compare the phase between pressure and motion to tell the direction of a sound source.

## Directional startle responses

To test whether *D. cerebrum* can hear sound direction, we tracked their motion in an aquarium surrounded by underwater speakers (Fig. 1c, Methods and Extended Data Fig. 1a,b). We played back transient sounds (about 12 ms duration, about 0.7 ms rise time, 780 Hz centre frequency; Methods and Extended Data Figs. 2 and 3) and quantified the direction of their startle reflex. The experiment was carried out with one fish at a time, and playback depended on the fish's orientation and position, to test left–right directional hearing and cancel position-dependent echoes (Fig. 1d and Methods).

Shortly after sound onset (within 17 ms or 2 video frames), *D. cerebrum* performed a characteristic startle reflex involving fast sideward displacement (see Fig. 1d,e, Methods and Extended Data Figs. 4 and 5 for startle dynamics, probabilities and habituation). We used the relative displacement along the left–right speaker axis 50 ms after startle onset as a readout of directional response and found that *D. cerebrum* startle away from the speaker, irrespective of the speaker location (left or right; Fig. 1f). Consequently, left and right responses were pooled in a metric for directional escape away from the speaker (Fig. 1g). These single-speaker sound playbacks resulted in directional escapes for 80% of all startles (191 of *n* = 240 startles across 63 fish with at least one startle, two-sided binomial test: *P* = 1 × 10⁻²¹), an effect present in both sexes (Extended Data Fig. 6a). A replication of the experiment in lateral line-ablated fish showed equivalent results of directional hearing (Extended Data Figs. 7b and 9b). Thus, female and male *D. cerebrum* exhibit directional hearing independent of lateral line function.

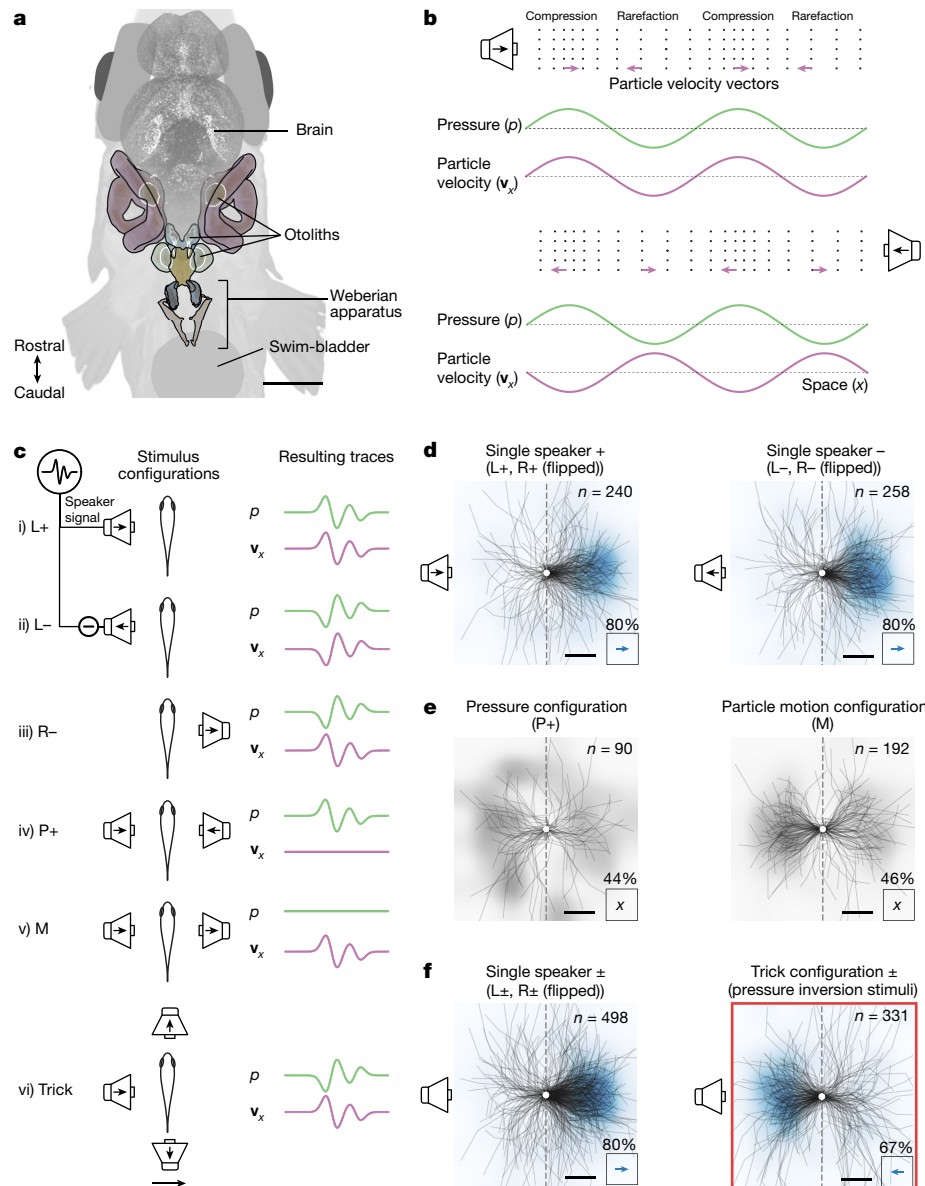

**Fig. 2 | The relative phase between pressure and particle motion predicts startle direction. a**, Schematic of *D. cerebrum* hearing apparatus anatomy. **b**, Schuijf's model: a plane wave from the left differs from a plane wave from the right in terms of the phase relationship between pressure and directed particle velocity. **c**, Illustration of stimulus configurations. Plus, minus and arrows inside speaker symbols refer to speaker signal polarity. L, left speaker; R, right speaker; P, only pressure; M, only motion. The schematics of resulting traces illustrate the pressure and particle velocity relationship (see Extended Data Figs. 2 and 3 for actual traces). Speaker schematics show a simplified configuration. Active echo cancellation typically involved three speakers (Supplementary Table 2). **d–f**, Centred startle trajectories after sound playback. All statistical tests are two-sided binomial tests. **d**, Directional

escapes away from a single speaker for a positive polarity signal (same data as Fig. 1g) and negative polarity signal (80% escapes, $P = 7 \times 10^{-23}$, $n = 258$ startle trials to this sound in 61 fish). **e**, Absence of significant directional bias in the positive polarity condition with only pressure (44% of $n = 90$ to right, not significant; $P > 0.05$) and only particle motion (46% of $n = 192$ to right, not significant; $P > 0.05$). **f**, Single-speaker playbacks pooled over both polarities evoke a directional escape (80% escapes, $P = 4 \times 10^{-43}$, $n = 498$ startle trials in 64 fish). Selective inversion of pressure polarity by an additional pair of speakers along the orthogonal axis inverts the relative polarity between pressure and particle velocity, which tricks the fish into performing startles approaching the active speaker (67% approach, $P = 1 \times 10^{-9}$, $n = 331$ startle trials in 61 fish). Scale bars, 500 µm (**a**) and 5 mm (**d–f**).

## Physical cues and hypotheses

We then inquired about possible algorithms that may underlie this directional hearing behaviour. As directionality has to be inferred from physical acoustic cues, we gathered those cues potentially available to *D. cerebrum* to list compatible hypotheses for a directional hearing algorithm. Adults have an inner ear separation (about 0.6 mm; Fig. 2a) that is orders of magnitude smaller than the wavelength of the sounds recorded in their natural habitat (≥150 mm for sounds up to 10 kHz). In addition, the mismatch between the characteristic acoustic impedance

of water (1.5 MPa s m$^{-1}$) and biological tissue (about 1.6–1.7 MPa s m$^{-1}$)[60] is small, unlike the approximately 4,000-fold mismatch between biological tissue and air (0.0004 MPa s m$^{-1}$). Hence, *D. cerebrum* can hardly break left–right symmetry by casting an acoustic shadow. However, close to a monopole sound source, ILDs could occur owing to the steep decay of the sound field with distance (Extended Data Fig. 8a). In *D. cerebrum*, the level drop between both inner ears at 3 cm distance from a monopole sound source can be as large as about 2% for pressure, irrespective of frequency, and about 4% for particle velocity at frequencies below 4.2 kHz. Hence, sound direction could be inferred

through two pressure sensors (P-ILD; hypothesis 1) or two particle motion sensors (M-ILD; hypothesis 2). These two strategies would work best for nearby sounds coming from a direction along the axis of the sensor pair.

The other binaural cue may be time differences in pressure (P-ITD, hypothesis 3) or particle motion (M-ITD, hypothesis 4). The maximal ITD between *D. cerebrum*'s inner ears is 0.4 µs, orders of magnitude smaller than in terrestrial vertebrates[4]. An ITD mechanism would thus point to an extreme sensitivity to minute time differences.

Each inner ear hair cell deflects along a preferred axis. Continuous sinusoidal sounds from opposing directions would stimulate the same hair cells, leading to the 180° ambiguity problem. If, however, all ecologically relevant sounds started with compression, the animal could interpret initial particle motion as motion directed away from the source. This sense would require just a single particle motion sensor for each axis. We call this possibility M-polarity hearing for either positive or negative polarity (hypotheses 5 and 6).

Finally, Schuijf's model for directional hearing resolves the 180° ambiguity problem by using pressure as a reference quantity (hypothesis 7). The idea is most easily illustrated for a plane wave (Fig. 2b), but its applicability is not limited to it (Extended Data Fig. 8b): at the phase of high compression, particles move away from the source, no matter if it started with push or pull. Thus, the pressure signal can act as a reference signal to resolve the 180° ambiguity of particle motion.

To distinguish between these hypotheses, we took advantage of *D. cerebrum*'s small size and precise control over stimuli under laboratory conditions. This allowed us to create stimuli that would normally not occur in nature (for example, sounds with pure pressure and no particle motion component) and to behaviourally dissect the algorithm used by *D. cerebrum* to tell sound direction.

## The directional hearing algorithm

The observed directional startle responses (Fig. 1f,g) are a reaction to naturalistic two-component sound consisting of pressure and particle motion signals (Fig. 2c(i)). To investigate how the isolated components of this sound field affect *D. cerebrum*, we created a pure pressure stimulus and a pure particle motion stimulus. The pure pressure stimulus was realized by driving opposing speakers in phase (Fig. 2c(iv)), and the pure motion stimulus was realized by driving them out of phase (Fig. 2c(v)), creating standing waves with nodes at the animal's location. Our approach additionally took into account echo cancellation and the spatially mapped frequency response of our recording tanks to present controlled and reproducible stimuli (Methods and Extended Data Figs. 2 and 3). We found that either component alone (pure pressure or pure motion) can elicit startles (Extended Data Fig. 7). However, neither component by itself elicited directionally biased responses, suggesting that both are necessary to tell sound direction (Fig. 2e and Extended Data Fig. 7).

The unbiased responses to the pure particle motion stimulus were the first evidence against a directional hearing algorithm based on initial particle motion polarity (M-polarity). To further test this hypothesis, we presented amplitude-inverted waveforms, for which the M-polarity algorithm would predict a reversed startle response. However, *D. cerebrum* performed startles away from the speaker for both polarities (Fig. 2d; pooled single-speaker data in Fig. 2f: 80% escape of $n = 498$ startles across 64 fish, two-sided binomial test: $P = 4 \times 10^{-43}$). Hence, we ruled out particle motion polarity (M-polarity) as the sole cue for directional hearing.

To test whether it may be the relationship between pressure and motion that determines the startle direction, we selectively inverted the pressure signal while leaving particle motion unchanged (similar to refs. 8,14). As we have seen, activating the left speaker (for example) evokes startles towards the right, away from the speaker, for both positive or negative waveform polarity (configurations shown

in Fig. 2c(i,ii)). By introducing an additional pressure source through in-phase activation of two speakers orthogonal to the left–right axis (Fig. 2c(vi)), we were able to selectively invert pressure, creating stimuli with pressure and particle motion cues akin to a sound originating from the right (Fig. 2c(iii)), despite the right speaker being inactive (Methods). In this 'trick condition', Schuijf's model predicts reversed startle behaviour (that is, 'escape' towards a speaker).

Indeed, *D. cerebrum* could be tricked: following pressure inversion, *D. cerebrum* performed startles towards the active speaker rather than away (Fig. 2f, 67% approach, two-sided binomial test: $P = 1 \times 10^{-9}$, $n = 331$ startle trials in 61 fish). This result held true for both sexes, within individual fish, for both pressure polarities and in lateral line-ablated fish (Extended Data Figs. 6b–d, 7 and 9d).

To check whether any binaural mechanism explains startles towards the speaker in the trick condition, we estimated the amplitudes and signs of P-ITD, M-ITD, P-ILD and M-ILD on the basis of theory and pressure measurements on both sides of the fish (see the section of the Methods entitled Estimation of binaural cues). We found that the signs of M-ITD and M-ILD remain unchanged when creating the trick condition and cannot explain the reversal of *D. cerebrum*'s escape direction. This suggests that neither interaural time nor level comparisons are sufficient to infer sound direction from particle motion.

In the single-speaker experiments, the distance-dependent decay of absolute pressure is 4.4 Pa over 600 µm, which is 2% of the pressure amplitude and potentially large enough to be detected. When we selectively invert pressure to realize the trick condition, we also effectively invert the sign of the level gradient along the horizontal $x$ axis, as well as the sign of the phase delay (see Extended Data Fig. 8d for a geometrical explanation). Therefore, P-ILD and P-ITD are the other mechanisms that are in agreement with the behavioural response of *D. cerebrum*.

In summary, *D. cerebrum* can be tricked into startling towards the speaker rather than away (see also Supplementary Video 1). Among all seven hypotheses considered, this behaviour is consistent with only three (Fig. 4): Schuijf's model, which relies on the phase comparison between pressure and particle motion; and the P-ILD and P-ITD mechanisms, which both rely on sensing pressure level at two positions in space.

To determine which of these remaining three hypotheses is compatible with *D. cerebrum*'s sensory anatomy, we next asked whether *D. cerebrum* possess sensory organs that can detect pressure and particle motion, as required by Schuijf's model, or sensory organs that can detect a difference in pressure amplitude along the azimuth, as required by the P-ILD and P-ITD mechanisms.

## The hearing apparatus

On the basis of micro-computed tomography (micro-CT) and optical vibrometry, we characterized *D. cerebrum*'s auditory organs to narrow down the candidate mechanisms for directional hearing. We started by visualizing the anatomy of *D. cerebrum*'s hearing apparatus with micro-CT imaging of a phosphomolybdic acid-stained sample (Methods). We segmented the main components of the hearing apparatus such as the swim bladder, the labyrinths, the otoliths (lapillus, asteriscus and sagitta) and otolithic end organs (utricle, lagena and saccule), the ossicles of the Weberian apparatus (including tripus and scaphium), and the lymphatic chambers that engulf the lagena and the saccule (Fig. 3a and Supplementary Video 2).

To study the motion of these auditory structures in response to the sound stimuli, we built an imaging vibrometer based on laser-scanning confocal reflectance microscopy: by time-gated acquisition of each pixel with respect to a continuous sinusoidal sound playback, it was possible to infer the relative phase of the motion of auditory structures in two dimensions (Fig. 3b, Methods and Extended Data Fig. 10).

First, we investigated whether the hearing apparatus of *D. cerebrum* is capable of detecting pressure. To this end, we drove two opposite

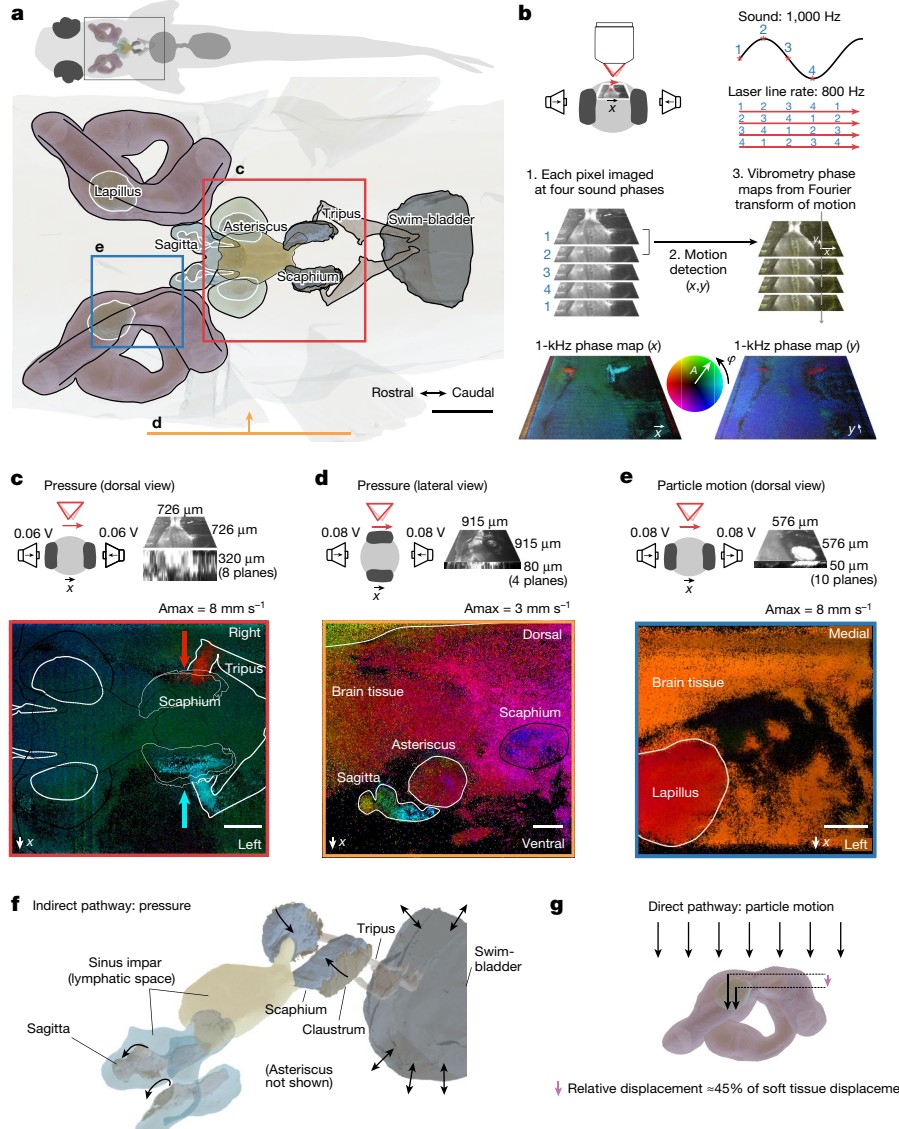

**Fig. 3 | The *D. cerebrum* hearing apparatus can detect pressure and particle motion separately. a**, Segmentation of the hearing apparatus based on micro-CT. The fields of view for vibrometry phase maps in **c**–**e** are indicated. **b**, Illustration of the vibrometry method. A laser-scanning confocal reflectance microscope is used to image the motion of auditory structures in a sound field. Each pixel is sampled at four sound phases to record sound-induced motion in the *x*–*y* plane. See Methods and Extended Data Fig. 10 for details. **c**–**e**, Particle velocity phase ($\varphi$) and amplitude ($A$) maps for motion along the left–right speaker axis; phase colour wheel shown in **b**, normalized to the amplitude $A_{max}$ indicated above the maps. **c**, A pressure stimulus causes anti-phase motion of the tips of the tripus and scaphium along the medial–lateral axis (indicated with red and cyan arrows). **d**, Along the dorso-ventral axis, a pressure stimulus creates phase-lagged motion of the scaphium and sagitta with the surrounding tissue. **e**, A particle motion stimulus results in relative motion between the

lapillus and surrounding tissue along the medial–lateral axis. Pooling over pixels within the region of interest, the displacement amplitude of the surrounding tissue is 1.24 µm ± 0.14 µm, and that of the lapillus is 0.95 µm ± 0.05 µm. At a relative phase of 0.14 π ± 0.05 π, the relative displacement can be as large as 0.56 µm (that is, 45% of the surrounding tissue motion). **f**, Indirect pathway for pressure sensing: in a pressure field, the swim bladder oscillates, which moves the tripus and the scaphium inwards and outwards. Lymphatic spaces probably couple the motion of the scaphium to the sagitta[73,74]. **g**, Direct pathway for particle motion sensing: the tissue of the fish couples to the particle motion of water, but the denser lapillus lags in phase and moves with lower amplitude, leading to a relative displacement that could be sensed by utricular hair cells underneath the lapillus. Observations shown in **c**–**e** were repeated once, four times and once in other fish, respectively, with similar results. Arrows in **f**,**g** indicate direction of motion. Scale bars, 250 µm (**a**) and 100 µm (**c**–**e**).

speakers in phase, thereby creating a 'pressure-only' stimulus. In this condition, we observed anti-phase motion of the left and right tripus and scaphium along the medial–lateral axis, periodically compressing lymphatic space (Fig. 3c,d). This motion stems from swim bladder compression and expansion in a pressure field. In addition, we observed rotational motion of the sagitta, part of the saccular end organ that lies embedded in an adjacent second lymphatic space (Fig. 3d). By contrast, we did not observe relative motion between the surrounding tissue and the lapillus or the asteriscus (Extended Data Fig. 11). We concluded that an indirect pressure sensing pathway exists in *D. cerebrum* and that

the saccule may be its main end organ, in agreement with findings in other fishes[38,39,61,62].

Second, we studied particle motion sensing by driving two opposite speakers in anti-phase. In this condition, with large particle motion but low-pressure signal, the lapillus, the otolith of the utricular end organ, moved with a phase lag (about 0.14 π ± 0.05 π) and at a lower amplitude (about 76% ± 10%) than the surrounding tissue (Fig. 3e). This creates a relative motion that is expected to stimulate the underlying hair cell epithelium (direct pathway). We did not detect such relative motion for the tripus, the sagitta and the asteriscus. Note that we used stimuli with

**Fig. 4 | Evidence for Schuijf's model of phase comparison.** We consider seven models for directional hearing that depend on different sensory structures and predict different behaviours (an eighth lateral line (LL)-based mechanism can be ruled out as we observe directional behaviour despite lateral line ablation). Interaural time difference (M-ITD) and level difference (M-ILD) mechanisms based on particle motion can be rejected on the basis of the behavioural data in the trick configuration. A strategy that is based on escaping positive (M-polarity (+)) or negative (M-polarity (−)) initial motion can be rejected as inverted polarity waveforms fail to invert the startle direction. Finally, a mechanism based on sensing pressure level or time differences (P-ILD, P-ITD)—consistent with behavioural data—can be rejected as *D. cerebrum* possess only a single pressure sensor. This leaves Schuijf's model as the one that correctly predicts an inversion of startle direction in the trick configuration and that is based on sensory cues that *D. cerebrum* is able to sense.

particle motion along the mediolateral axis relevant for the left–right startle behaviour. To support directional hearing in three dimensions, *D. cerebrum* may have further direct motion sensing pathways along additional axes, in line with particle motion tuning in saccular[63,64] or lagenar[63,65] afferents in other species. In summary, we did not find two pressure sensors that could detect a pressure level difference (P-ILD) or time difference (P-ITD) along the binaural axis, but rather a single pressure sensor (Fig. 3f) and a set of particle motion sensors (Fig. 3g).

We consider whether *D. cerebrum* could have another pressure sensor that the vibrometry measurements did not detect. Like in other cyprinids, their swim bladder is divided into an anterior and a posterior part. To rule out *D. cerebrum* using a pressure difference between these divisions, we repeated our behavioural analysis for only those startles in which the anterior–posterior axis of the fish was near-orthogonal to the axis of sound presentation, giving equivalent results (Extended Data Fig. 12a). Theoretically, *D. cerebrum* might possess other, unknown pressure sensors to implement P-ILD or P-ITD. However, all known sound pressure sensors in fish are based on compressible gas-filled organs. As gas-filled structures have a high micro-CT contrast, hypothetical pressure sensing organs would have to be either microscopic to evade detection, or based on an unknown principle of sound pressure transduction without compressible gas. Neither of these options is supported by our current knowledge of fish biology and physics of sound.

We therefore reject P-ILD and P-ITD as plausible mechanisms. Instead, the *D. cerebrum* anatomy is well suited for implementing Schuijf's model for directional hearing.

## Discussion

In this study, we report direct experimental evidence for directional hearing in fish and identify its underlying biophysical mechanism. Directional startles were acoustically elicited in male and female *D. cerebrum*, irrespective of lateral line ablation. A selective inversion of

the pressure component tricked *D. cerebrum* into performing a startle towards the active speaker rather than away, consistent with the hypothesis that fish compare pressure and particle motion signals to infer direction (Schuijf's model). On the basis of anatomical and behavioural data, we rejected all known alternative models, including binaural models for directional hearing. Using optical vibrometry, we confirmed the existence of a direct pathway for particle motion sensing and an indirect pathway for pressure sensing in *D. cerebrum*. Hence, the *D. cerebrum* hearing apparatus supports a dual sense of hearing that allows for a comparison between pressure and particle motion. Together, these findings suggest that Schuijf's model is actually implemented in nature (Fig. 4).

Our work builds on a large body of pioneering work on fish hearing, summarized in Supplementary Table 1. To provide an overview of past evidence for directional hearing in fish, we categorized publications into five study types. We indicated to what degree authors controlled acoustic variables and whether they could judge their importance to directional hearing. These previous studies faced a trade-off: they were carried out either in open water, in which reverberations are negligible, but experiments are challenging so that only a few fish were tested[66], or in the laboratory, in which more fish could be tested but pressure and particle motion were not fully controlled, and the lateral line function could not be ruled out as a near-field sense that aids directional hearing. Here we addressed these limitations by measuring the impulse response of our speakers, actively cancelling reverberations and precisely controlling natural and unnatural stimuli.

Hearing in humans refers to the perception of pressure oscillations. We have shown that *D. cerebrum* instead have a dual sense of hearing comprising pressure and particle motion sensing, which is used for directional hearing. It could be that the acoustic world is much richer for fish than for humans, with acoustic events carrying stereotypic dual pressure-motion signatures[67,68] that may even reveal their distance[69]. As *D. cerebrum* is a vocal species, directional hearing may also have a social function.

Schuijf's model has recently been extended through a proposal that fish compute the time-averaged product of pressure and motion (the acoustic intensity vector)[6]. This theory can account for phonotaxis of plainfin midshipmen towards monopole and dipole sources[17,18]. Future neurophysiological work may test whether *D. cerebrum* implements Schuijf's model this way.

*D. cerebrum* shares the Weberian apparatus with other otophysans, a superorder that includes 66% of living freshwater fish species and 15% of all living vertebrate species[33,70]. Even in fishes lacking the Weberian apparatus, otolithic end organs may still inherit pressure-induced swim bladder motion[38,62,71,72]. Hence, Schuijf's model may have widespread applicability.

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

# Methods

## Animals

All animal experiments conformed to Berlin state, German federal and European Union animal welfare regulations and were approved by the LAGeSo, the Berlin authority for animal experiments. *D. cerebrum* were kept in commercial zebrafish aquaria (Tecniplast) with the following water parameters: pH 7.3, conductivity 350 µS cm⁻¹, temperature 27 °C. We used male and female adult fish between 4 and 11 months of age.

## Behavioural setup and protocol

The experimental setup comprised an inner 10 cm × 10 cm (length × width) tank with <200 µm thin optically opaque but acoustically transparent polypropylene sheet walls (cut out of plastic folders), surrounded by an outer tank with submerged speakers (4 × 3 Ekulit LSF-27M/SC 8 Ω in custom waterproof enclosures). Thus, the speakers were visually shielded from the fish inside the inner tank, and the fish were confined between the speakers (Fig. 1c provides a schematic; further details are provided in Extended Data Fig. 1a,b). The height of the water was 10 cm, and the transparent bottom of the inner tank was at 6.3 cm, leaving 3.7 cm to the water surface as a water column for the fish to swim in. The speakers were level with this water column, and all sounds were targeted for this water column. Infrared light-emitting diodes illuminated the fish from below. The inner tank was filmed with an overhead camera at 120 fps at 336 × 336 pixel resolution, and live tracking of the fish was carried out on a subset of frames at 15 fps. White light-emitting diodes lit the setup indirectly via reflections from the room walls. The room and water temperature was kept at 27 °C.

Each fish was tested once, and one fish was tested at a time. In the first minutes of the recording, a 10 cm × 10 cm acrylic plate with centimetre markings was placed in the inner tank to match the sound calibration grid to the video frame. Three minutes after placing the fish in the inner tank, playbacks were triggered for 45 min.

To probe into left–right directional hearing, playbacks from the front or back of the fish should be avoided. We prompted *D. cerebrum* to align with respect to the left–right speaker *x* axis to increase experimental throughput: previously, we observed that *D. cerebrum* swim closer to white than to black walls. By using two black plastic films as walls across the *x* axis and two white films across the *y* axis, we encouraged *D. cerebrum* to oscillate between the white walls, along the *y* axis (Extended Data Fig. 1c). Consequently, the ratio of distance covered along the *y* axis to the distance covered along the *x* axis was 1.6. Sound playback occurred only when fish were orthogonally oriented within a 45° angle measured from the orthogonal axis and within a 1.5 cm × 3 cm trigger zone at the centre, leaving at least 3.5 cm distance to the nearest wall (the typical startle displacement is mean ± s.d. = 1 cm ± 0.4 cm after the first 50 ms). The minimal delay between playbacks was set to ≥5 s with a minimum delay of 5 s plus a random delay, drawn from an exponential distribution with a mean of 5 s for each trial. This paradigm averaged to about 1.3 playbacks per minute in untreated fish and to about 0.6 playbacks per minute in lateral line-ablated fish.

Twelve target sounds were generated from a recorded pressure waveform (see the section of the Methods entitled Sound stimulation waveforms), targeted to the fish's current position to cancel reverberations (see the section of the Methods entitled Calibration and reverberation cancellation), and presented to the fish in random order following trigger events using custom-written code in Python 3.

The data in Figs. 1–4 and Extended Data Figs. 4–7 and 12a stem from 65 untreated fish (3,798 playbacks, 1,415 startles, about 37% startles). For each stimulus, we indicated the number of fish that responded with at least one startle. The same experiment, also comprising 12 sound configurations, was repeated with 74 lateral line-ablated fish (Extended Data Figs. 7 and 9; 2013 playbacks, 910 startles, 45% startles). A third sound playback experiment was carried out in the dark in 43 untreated fish, testing a subset of 4 sound configurations (Extended Data Fig. 12b).

## Behavioural analysis

**Tracking.** Pose tracking of *D. cerebrum*'s swimming behaviour was carried out with SLEAP[75]. In total, 140 frames across nine random recordings of male and female fish were hand-labelled with a skeleton consisting of 7 equidistant nodes along the fish's body segments and 2 additional nodes, 1 for each eye. The 'single-animal' model was used for training. The model parameters and the trained model are available at the G-Node repository (see Data availability).

**Startle detection.** Plotting the fish's velocity against time around playback revealed a sharp increase in velocity after playback, clearly visible across all playbacks (Fig. 1e and Extended Data Fig. 4a). We defined a 25-ms time window around the time of peak velocity at which the speed distribution is bimodal and computed the average velocity in this time window for each trial to classify all playback trials with an average velocity above 17 cm s⁻¹ as startle trials (Extended Data Fig. 4b,c). The remaining ones were classified as non-startle trials. The decision criterion based on speed also resulted in a clear separation in terms of body bend (Extended Data Fig. 4d).

**Directional bias.** To classify startles into left or right, we measured the fish's *x* displacement during the first 50 ms after startle initiation (Fig. 1e). This duration was chosen because displacement heat maps at varying delays revealed that the initial, lateral displacement phase of the startle response peaks after 50 ms (Extended Data Fig. 4f). Wherever centred trajectories are shown, these initial 50 ms are depicted. The directional bias of the startle response is the fraction of startles to one indicated direction (left or right, away or towards speaker). This bias can be computed in two ways.

*Directional bias across trials and fish.* For each stimulus or set of stimuli, startle trials were pooled across all fish, and the fraction of startles in one direction was calculated. Using the two-sided binomial test, we calculated how likely a measured directional bias (approach or escape) would have been observed if the response was unbiased.

*Directional bias per fish.* In the analysis of bias across trials and fish, theoretically, all trials could stem from one performing animal (not of concern here; Extended Data Fig. 5a,b). To complement this measure, we also quantified the directional bias per fish. We had 12 sound configurations in each experiment and startles averaged to a total of about 22 startles per fish in an experiment; hence, a meaningful per-fish bias could be computed only on pooled sound configurations and for fish with many startles. To estimate the directional bias of individual fish, we filtered for fish that had ≥10 startles in both the single-speaker condition (pooled over 4 stimuli) and the trick condition (pooled over 4 stimuli; Extended Data Fig. 6c,d). Although the value reflects directional behaviour in the population and estimates fish-to-fish variability, it selects for fish that trigger many playbacks and startle often.

## Micro-CT

A 12-month-old male wild-type *D. cerebrum* was euthanized by ice shock and fixed with 4% paraformaldehyde in phosphate-buffered saline (PBS) at 4 °C overnight. The next day, the fish was washed for 15 min in PBS before being stained with 5% phosphomolybdic acid (Sigma Aldrich) solution in PBS at 4 °C overnight. After staining, the fish was washed in PBS for 15 min before embedding in 1% PBS-buffered agarose inside a cryo tube. The micro-CT scan was carried out at the ANATOMIX beamline at SOLEIL synchrotron by XPLORAYTION. The sample was placed into a 40-keV polychromatic (white) X-ray beam. A scan consisted of 3,200 projections collected at about ×10 optical magnification by a digital camera (Orca Flash 4.0 V2) with a sensor pixel size of 6.5 µm at 150 ms exposure time, yielding an effective pixel size of 0.6485 µm. The registered data were binned to 1.2970 µm voxel size. Key structures of the hearing apparatus were manually segmented. To this end, planes were hand-labelled using 3D Slicer[76] (v5.6, https://

slicer.org) and then interpolated using Biomedisa (v23)[77]. FIJI ImageJ (v1.5)[78] was used to convert between different file types. The segments were turned into mesh grids and loaded into Blender for cleaning and rendering.

### Lateral line ablation and DASPEI staining

To rule out that the lateral line organ senses sound directionality in our experiments, we ablated the lateral line using neomycin[79]. To ablate the neuromasts, fish were placed in a 200 µM neomycin solution for about 30 min. Afterwards, they were transferred to a beaker with tank water. Behavioural experiments started after ≥30 min. To confirm the reliability of the lateral line ablation protocol, we stained 30 neomycin-treated fish. After the behaviour experiment, they were transferred to a 100 µM DASPEI (2-[4-(dimethylamino)styryl]-1-ethylpyridinium iodide) solution and then to a beaker with tank water to wash out unbound DASPEI. Afterwards, the fish were euthanized with an ice shock and imaged with an epifluorescence microscope. Neuromasts were reliably stained in control fish but not in neomycin-treated fish, indicating reliable ablation (see Extended Data Fig. 9e,f for example images). As functional metrics we report an increase in number of wall contacts after startles (Extended Data Fig. 9g) and a decrease in foraging strikes in the dark (Extended Data Fig. 9h) in neomycin-treated fish.

### Vibrometry

**Confocal microscope.** The confocal reflectance microscope was based on a custom-built laser-scanning two-photon microscope (Extended Data Fig. 10a). The illumination source was a Ti:sapphire laser (MaiTai DeepSee; SpectraPhysics) operated at 810 nm (with or without mode-locking). Before entering a laser-scanning two-photon microscope, the beam passed through a 90:10 beam splitter (90% reflection, 10% transmission). The light back-scattered by the fish inner structures was descanned, reflected by the 90:10 beam splitter, and then focused by a lens ($f$ = 50 mm) into a single-mode fibre (core diameter: 25 µm, numerical aperture: 0.1) acting as a confocal pinhole. The microscope was controlled by custom-written software (https://github.com/danionella/lsmaq).

**Acoustic stimulation.** Fish were anaesthetized in 120 mg l$^{-1}$ fish water-buffered MS-222. They were subsequently placed on a preformed agarose mould, which allowed the gill covers to move freely, and immobilized with 2% low-melting-point agarose (melting point 25 °C). A flow of aerated aquarium water (with anaesthetic) was delivered to their mouth through a glass capillary.

The fish was acoustically stimulated using two facing speakers sealed in custom-made waterproof enclosures. The diaphragms were exposed to water. The speakers were each placed about 1.3 cm away from the fish. They were driven using a DAQ card (National Instruments USB-6211), connected through audio amplifiers (Kemo M031N, 3.5 W). Pressures of up to about 176 dB (referenced to 1 µPa) were thus generated at the fish position in the pressure-only configuration and particle motion of up to about 8 mm s$^{-1}$ in the particle-motion-only configuration, consistent with the expected amplitude relationship between pressure and particle motion in the sound monopole near field.

**Motion phase maps.** The principle of the laser-scanning vibrometric measurement is illustrated in Fig. 3b and Extended Data Fig. 10b. The sample (Extended Data Fig. 10b(i)) was stimulated with an acoustic sinusoidal wave at frequency $f_{stim}$, and imaged with a laser-scanning microscope with a line rate $f_{scan}$ (Extended Data Fig. 10b(ii)).

To reconstruct amplitudes and relative phases of sinusoidal object motion, we needed to measure each pixel under more than two different phases according to the Shannon–Nyquist sampling theorem. As noise can influence this measurement, we used four phase steps here, ensuring proper phase reconstruction while keeping acquisition sessions reasonably short.

To reconstruct the displacement of the moving structures inside the fish, each line of the image was repeatedly scanned nStep = 4 times, with a phase offset of π/2 between each line (Extended Data Fig. 10b(iii)). To this end, the stimulation frequency and the line rate must follow the relationship:

$$f_{stim} = (N + 1/nStep)f_{scan}$$

with $N$ being an integer. To maximize the line rate, we took $N = \text{floor}(f_{stim}/f_{scan})$.

This in turn set additional constraints on the various scanning parameters. We used $f_{scan}$ = 800 Hz and $f_{stim}$ = 1,000 Hz for the data presented in Fig. 3 and Extended Data Fig. 11.

To ensure repeatable measurements, the acoustic stimulation and the galvanometric scanning mirrors were synchronized so that each pixel was recorded at a known sound phase. This was achieved by triggering the sound generation on each single frame scan trigger.

Doing so, each pixel was stroboscopically probed at nStep = 4 different phases of the acoustic stimulation cycle. As sound propagates while scanning two consecutive pixels, the probed acoustic phase is shifted by 2π × pixelPeriod, which was taken into account in the motion reconstruction of the imaged structures (Extended Data Fig. 10c). These images were then reshaped to yield an (Nx,Ny,nStep) dataset (Extended Data Fig. 10b(iv)).

To analyse the motion of the inner structures of the fish, we used Matlab 2019b and a particle image velocimetry toolbox PIVlab[80], originally developed to characterize the motion of flowing particles for fluid mechanics. Essentially, the particle displacement is assessed by cross-correlating subregions with decreasing sizes of consecutive images (Extended Data Fig. 10b(v)). The contrast of the reflectance images was enhanced before the displacement analysis, and the results were curated in post-processing by removing outliers and interpolating detection gaps.

The motion detection yielded $x$- and $y$-displacement maps at each of the four phases in the acoustic stimulation period. The first Fourier component was computed for each pixel to extract the amplitude and phase of the local displacement (Extended Data Fig. 10b(vi)). The phase was finally corrected for the accumulating phase offset along the horizontal $x$ direction due to the line scanning procedure (Extended Data Fig. 10c). Owing to the synchronization of the acoustic stimulation with the line scanning process, we could carry out this measurement in several planes and obtain a consistent volumetric complex map characterizing the motion response of the various inner structures to the acoustic stimulation. Maximum-amplitude projections across planes delivered the shown two-dimensional phase maps, one for motion along the speaker–speaker axis ($x$) and one for motion orthogonal to the speaker–speaker axis ($y$).

### Sound stimulation waveforms

We reasoned that *D. cerebrum* sense pressure and particle motion. Hence, our sound stimuli were defined in terms of three quantities: pressure, $x$ acceleration and $y$ acceleration, which were delivered to the fish's current position by utilizing the frequency responses of speakers to cancel position-dependent reverberations (see the section of the Methods entitled Calibration and reverberation cancellation). $y$ acceleration was always kept at zero, and only pressure and $x$ acceleration were varied. In summary, 12 sounds were generated from a recorded pressure waveform and presented to the fish in a random sequence upon trigger events. The 12 sounds consisted of four single-speaker sounds (left or right × positive polarity or negative polarity), two sounds with only a pressure component (positive polarity or negative polarity), two sounds with only horizontal $x$-motion components (positive polarity or negative polarity) and four trick conditions, which exactly matched the four single-speaker target waveforms, but differed by the speakers that were active to realize these.

We observed that *D. cerebrum* startle when we drop a cylindrical piece of rubber into the water. We recorded the pressure waveform of this sound, high-pass filtered it at 100 Hz, and extracted a 12-ms snippet to serve as our pressure waveform template (note that conditioned sounds—that is, the actual speaker signals—were band-pass-filtered between 200 Hz and 1,200 Hz; see the following section). The target pressure amplitude was set to a peak sound pressure level of 167 dB (referenced to 1 µPa) by rescaling this pressure waveform accordingly. This amplitude was loud enough to elicit startles reliably and still supported by our small 2.7-cm-diameter speakers. The first peak's rise time (10% to 90% absolute amplitude) was 0.664 ms and the centre frequency of the pulse was about 780 Hz. The target horizontal particle acceleration waveform was computed from the pressure waveform using monopole theory for each Fourier component, as follows.

The pressure signal decays as $1/r$ with radial distance $r$ away from a sound monopole with amplitude $p_0$ at distance $r_0$

$$\hat{p}(r,t) = \hat{p}_0 \frac{r_0 e^{ikr}}{r} e^{-i\omega t}$$

and with frequency $f$, $\omega = 2\pi f$, wavenumber $k = 2\pi/\lambda$, wavelength $\lambda$ and speed of sound $c = \lambda f$.

In a medium of density $\rho$, the radial particle velocity decays quadratically with distance in the near field ($kr \ll 1$, limit dependent on frequency):

$$\hat{v}_r(r,t) = \left[ \frac{1}{\rho c}\left(1 + \frac{i}{kr}\right)\hat{p}(r,t)\right]$$

By contrast, particle acceleration—the temporal derivative of particle velocity—decays quadratically with distance for nearby sounds ($r \ll 1$, limit independent of frequency):

$$\hat{a}_r(r,t) = -i\omega \hat{v}_r(r,t) = \frac{1}{\rho}\left(\frac{1}{r} - ik\right)\hat{p}(r,t)$$

To compute the particle acceleration $a_r(r,t)$ at a distance $r$ to a sound monopole with pressure $p(r,t)$ for discrete signals of arbitrary waveform, we applied this equation separately for each Fourier component. Given a pressure waveform $\{\mathbf{p}_n\} := p_0, p_1, \cdots, p_{N-1}$ with $N$ samples $p_n$, spaced at $T = 1/sr$ with sample rate $sr$, the particle acceleration $\{\mathbf{a}_n\} := a_0, a_1, \cdots, a_{N-1}$ that would be observed at a distance $r = r_0$ from a sound monopole was calculated by carrying out the discrete Fourier transform $\{\mathbf{P}_l\} := P_0, P_1, \cdots, P_{N-1}$

$$P_l = \sum_{n=0}^{N-1} p_n e^{-\frac{i2\pi}{N}ln}$$

and deriving particle acceleration for each Fourier component $\{\mathbf{A}_l\} := A_0, A_1, \cdots, A_{N-1}$ independently. With corresponding frequencies $f_l \approx l/(NT)$, such that $k \approx 2\pi l/(NTc)$, and the relationship between pressure and particle acceleration, $A_l$, is calculated as

$$A_l = \frac{1}{\rho}\left(\frac{1}{r_0} - ik_l\right)P_l$$

which defines the radial particle acceleration through the inverse Fourier transform:

$$a_n = \frac{1}{N}\sum_{l=0}^{N-1} A_l \, e^{i\frac{2\pi}{N}ln}$$

In the experiments, $r_0$ was set to 3 cm, thus simulating a monopole sound source at 3 cm, irrespective of *D. cerebrum*'s relative position to the speakers. This resulted in a peak particle acceleration of 7.59 m s$^{-2}$.

Other parameters used were $c = 1,500$ m s$^{-1}$, $\rho = 1,000$ kg m$^{-3}$ and sr = 51,200 Hz. In terms of pressure, $x$ acceleration and $y$ acceleration ($p$, $a_x$ and $a_y$), there were eight different target configurations, with '+' indicating polarity of the template waveform and '−' indicating opposite polarity: four monopole configurations (+,+,0), (−,−,0), (+,−,0) and (−,+,0); two pressure configurations (+,0,0) and (−,0,0); and two motion configurations (0,+,0) and (0,−,0). Despite a total of eight target configurations, there were 12 sound configurations as the four monopole configurations can be realized in two ways, either with a single speaker or with three speakers (trick configuration; see the next section).

### Calibration and reverberation cancellation

Conducting experiments in small tanks presents challenges as both tank geometry and the receiver's position affect the sound amplitude and waveform sensed by the receiver (Extended Data Fig. 2c). By recording the speakers' impulse responses inside the inner tank in terms of pressure and particle acceleration (Extended Data Fig. 2b), speakers could be activated to precisely control pressure and particle acceleration components at the fish's location (Extended Data Fig. 2d).

**Pressure and acceleration measurements.** *Pressure.* Pressure was measured with a hydrophone (Aquarian Scientific AS-1, preamplifier: Aquarian Scientific PA-4, acquisition: NI-9231 sound and vibration module, National Instruments; Extended Data Fig. 1d). During repeated playback of the same sound, a single hydrophone was automatically moved across a 5 × 5 grid inside the inner tank, sampling with a spacing of 1.5 cm (Extended Data Fig. 1c,e). Hence, a 25-point pressure field was obtained for each sound configuration, spanning a 6 cm × 6 cm square at the centre of the inner tank between the speakers.
*Acceleration.* Particle acceleration was measured in two ways.

In the first method, particle acceleration was measured indirectly through the pressure gradient. Newton's second law of motion (pressure gradient force)

$$\mathbf{a} = -\frac{1}{\rho}\nabla P$$

links the spatial pressure gradient to particle acceleration. In water, with density $\rho = 1,000$ kg m$^{-3}$ and speed of sound $c = 1,500$ m s$^{-1}$, the following approximation holds for pressure signal frequencies $f \ll 100$ kHz, if the pressure gradient is sampled with step size $x_2 - x_1 = 1.5$ cm:

$$a_x = -\frac{1}{\rho}\frac{p(x_2) - p(x_1)}{x_2 - x_1}$$

The approximation holds for all frequencies used in this experiment. For measuring gradients, moving a single hydrophone is preferred over a hydrophone array, as the gradient could be biased by small differences in hydrophone sensitivity and perturbations of the sound field by the presence of other hydrophones. We calculated $x$ and $y$ acceleration on the basis of the 25-point pressure field recorded with a single hydrophone. The pressure field included points outside the trigger zone to compute pressure gradients (that is, acceleration) across the trigger zone boundary.

In the second method, particle acceleration was additionally directly measured along all three axes with an acceleration sensor (Triaxial ICP - Model 356A45, PCB Piezotronics, acquired with NI-9231 sound and vibration module, National Instruments; Extended Data Fig. 1d). Like the hydrophone, the acceleration sensor was moved across all 5 × 5 grid positions during repeated playback of the same sound, giving measurements for $x$, $y$ and $z$ acceleration.

Whereas hydrophones are manufactured and calibrated for underwater use, the particle acceleration sensor is not made to measure particle acceleration underwater and is meant to be glued onto the

vibrating object. Owing to an acoustic impedance mismatch between metal and water, we expected the PCB sensor to underestimate particle acceleration.

We compared $x$ and $y$ acceleration waveforms for both measurement methods and found that the acceleration waveforms acquired through the direct method match the waveforms acquired through the indirect method after multiplication by a factor of about 2.4. The close match in rescaled waveforms confirms the validity of the gradient approximation in the indirect method.

Hence, in all experiments, $x$ and $y$ acceleration were measured through the indirect method, on the basis of spatial pressure gradients. The particle acceleration sensor still proved useful in measuring the vertical $z$ acceleration in our setup.

**Impulse response-based sound targeting.** To create the same sounds at any position inside the inner tank, impulse responses for all 4 speakers were measured across 25 positions on a 5 × 5 grid with 1.5-cm spacing. In the following, the sound targeting method is described for one position.

Let $k_{i,p}$ be the pressure impulse response kernel, $k_{i,a_x}$ be the $x$ acceleration impulse response kernel, and $k_{i,a_y}$ be the $y$ acceleration impulse response kernel for the $i$th speaker. Using $M$ speakers with signal $s_i$, pressure and acceleration can be predicted through convolution ($*$):

$$p = \sum_{i=0}^{M-1} k_{i,p} * s_i$$

$$a_x = \sum_{i=0}^{M-1} k_{i,a_x} * s_i$$

$$a_y = \sum_{i=0}^{M-1} k_{i,a_y} * s_i$$

In the Fourier domain, utilizing the convolution theorem, these become a system of equations for each Fourier component $l$.

$$P_l = \sum_{i=0}^{M-1} K_{i,p,l} S_{i,l}$$

$$A_{x,l} = \sum_{i=0}^{M-1} K_{i,a_x,l} S_{i,l}$$

$$A_{y,l} = \sum_{i=0}^{M-1} K_{i,a_y,l} S_{i,l}$$

On the basis of the Fourier components of the target waveforms (see the section of the Methods entitled Sound stimulation waveforms), $P_l$, $A_{x,l}$ and $A_{y,l}$, and the Fourier components of the impulse response kernel $K_{i,p,l}$, $K_{i,a_x,l}$ and $K_{i,a_y,l}$, the system of equations can be solved for the Fourier components of the speaker signals $S_{i,l}$ as long as $M \geq 3$ and the kernel components are non-zero and non-identical. The time-domain signal for the $i$th speaker is then given by the inverse Fourier transform using components $S_{i,l}$.

To increase robustness of the solutions (for example, to avoid speakers cancelling themselves unnecessarily and to limit speaker amplitude), speaker signal waveforms were forced to become similar to the target waveform. This was implemented by solving the system of equations with a least-square solver (scipy.optimize.lsq_linear) with bounds $-B_{i,l} < S_{i,l} < B_{i,l}$. The bound $B_{i,l}$ was computed as a rescaling of the absolute Fourier components of the target pressure waveform $P_l$

$$B_l = \alpha_i \gamma |P_l|$$

in which $\gamma$ is fixed and scales pressure to voltage and $\alpha_i$ is a rescaling parameter set independently for each speaker to give additional control over active speakers. We list our values for $\alpha_i$ used in different sound configurations in Supplementary Table 2.

After conditioning, all computed speaker signals were band-pass-filtered between 200 Hz and 1,200 Hz to avoid activating the lateral line.

To ensure that the trick configuration differed from the single-speaker configuration only by selective pressure inversion, a two-step sound conditioning was carried out. First, the speaker signals for the single-speaker configuration were calculated. Then, these signals were effectively fixed to closely resemble the single-speaker signal and only activations of the two speakers along the orthogonal axis were conditioned.

The above calculation was carried out for the 25 grid positions. The computed speaker signals accurately delivered the target waveforms to the target position (Extended Data Figs. 2d and 3 and Supplementary Table 2). To ensure consistency over experiments, the water level was kept at precisely 10 cm, and the pressure and acceleration fields inside the inner tank were measured several times (this includes before the first recording and after the last recording).

During the experiment, the fish's $x$–$y$ position was detected at 15 Hz, and the loading for the speakers was linearly interpolated on the basis of targeted sounds at neighbouring grid positions.

In the section entitled Sound stimulation waveforms, we describe how we defined the pressure and particle motion target waveforms that were conditioned this way.

### Estimation of binaural cues (P-ILD, M-ILD, P-ITD, M-ITD)

**ILDs.** To estimate binaural cues in our behavioural experiment, we analysed the pressure and particle motion at sound calibration grid points 3 cm apart, ($x_0$) 1.5 cm to the left and ($x_1$) 1.5 cm to the right of the centre grid point. To estimate sign and peak amplitude of level differences, we calculated P-ILD (pressure ILD) as $\max_t(\mathrm{abs}(p(x_0, t))) - \max_t(\mathrm{abs}(p(x_1, t)))$ and M-ILD (particle motion ILD) as $\max_t(\mathrm{abs}(a_x(x_0, t))) - \max_t(\mathrm{abs}(a_x(x_1, t)))$. The level differences between these two points were divided by a factor 50 to estimate the level difference across the left-to-right inner ear axis of the fish (about 0.6 mm). Comparing the single-speaker configuration with the trick configuration, these data show that the sign of M-ILD remains the same (+0.11 m s$^{-2}$ versus +0.30 m s$^{-2}$), but the sign of P-ILD is inverted (+4.4 Pa versus −4.6 Pa). For a geometrical illustration of the inversion of P-ILD, see Extended Data Fig. 8d.

**ITDs.** ITDs were estimated by calculating the phase propagation in different sound configurations under a monopole approximation (Extended Data Fig. 8c–f).

### Reporting summary

Further information on research design is available in the Nature Portfolio Reporting Summary linked to this article.

## Data availability

Fish trajectory, micro-CT and vibrometry data have been deposited in the G-Node repository at https://gin.g-node.org/danionella/veith_et_al_2024. Downsampled segmentation mesh-grid data have been deposited to MorphoSource at https://n2t.net/ark:/87602/m4/623999. Source data are provided with this paper.

## Code availability

Code used for data analysis is available on GitHub at https://github.com/danionella/veith_et_al_2024.

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

**Acknowledgements** We thank M. Brecht, M. Larkum, J. Poulet, M. Lovett-Barron, T. Veith, B. Hesse, D. Markov, M. Kadobianskyi, V. Cook, L. Constien, J. Henninger, A. Groneberg, U. Böhm, L. Kosina, F. O'Donovan, L. Schulze and J. Tiefenbacher for discussions and for critically reading our manuscript; and our fish facility team, A. Wrana, M. Renz, D. Krap, H. Schmidt, N. Kroworz, for fish care and experimental support. We acknowledge support by the European Research Council (ERC2021-CoG-101043615), the Einstein Foundation (EPP-2017-413), the German Research Foundation (DFG, project EXC-2049-390688087) and the Alfried Krupp von Bohlen und Halbach Foundation. T.C. received a fellowship from the Alexander von Humboldt Foundation.

**Author contributions** Conceptualization: J.V., T.C. and B.J. Methodology: J.V., T.C., M.H. and B.G. Investigation: J.V., T.C., A.S., L.E.D. and M.H. Visualization: J.V. and T.C. Funding acquisition: B.J. Project administration: B.J. Supervision: T.C. and B.J. Writing: J.V., T.C. and B.J.

**Competing interests** The authors declare no competing interests.

**Additional information**
**Correspondence and requests for materials** should be addressed to Benjamin Judkewitz.

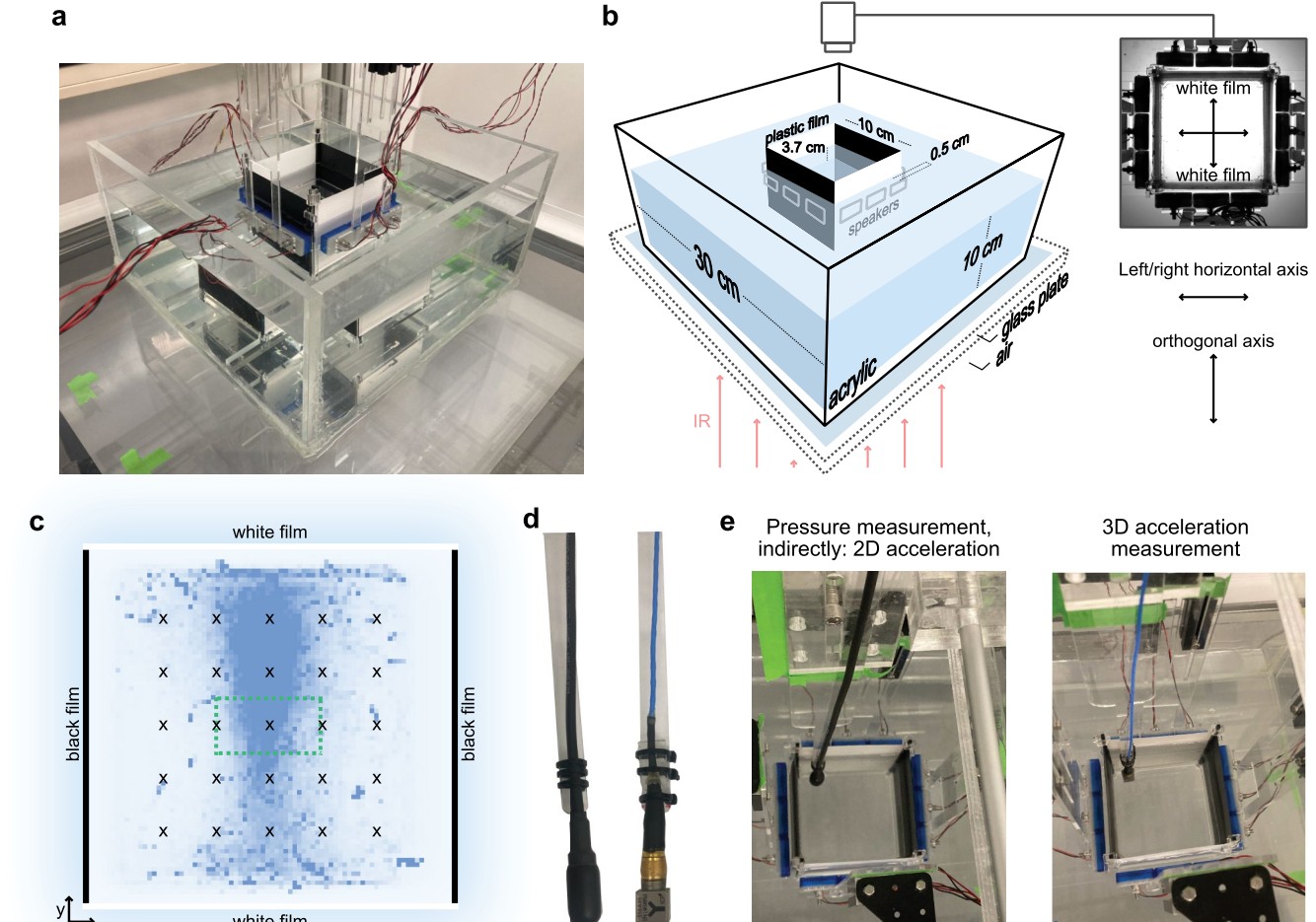

**Extended Data Fig. 1 | Setup design and sound calibration equipment.**
**a**, Photo of the setup. **b**, Schematic of the setup indicating size and material.
The room and water temperature is kept at 27 °C; the room light is indirect light
reflecting from white walls. Infrared (IR) light illuminates the setup from below.
**c**, Schematic of sound calibration points inside the inner tank overlaid over
a heatmap of fish positions inside the inner tank across all 65 recordings.
*D. cerebrum* avoids the black walls and oscillates between the two white walls.
Pressure and particle acceleration were measured at 25 points (x) on a grid
spanning a 6 cm x 6 cm square, covering the trigger zone (green dotted

rectangle). Sounds were conditioned to deliver target waveforms to the fish's
swimming position within the trigger zone based on each speaker's pressure
and acceleration impulse responses for these 25 points. Points outside the
trigger zone were measured to calculate pressure gradients across the edge of
the trigger zone, see Methods. **d**, Left: Hydrophone, Aquarian Scientific, AS-1.
Right: Triaxial acceleration sensor, ICP® - Model 356A45, PCB Piezotronics. **e**,
The motorized arm moved the sensor to each grid position. Left: pressure
measurement (was also used to compute acceleration indirectly). Right: direct
acceleration measurement with the PCB sensor.

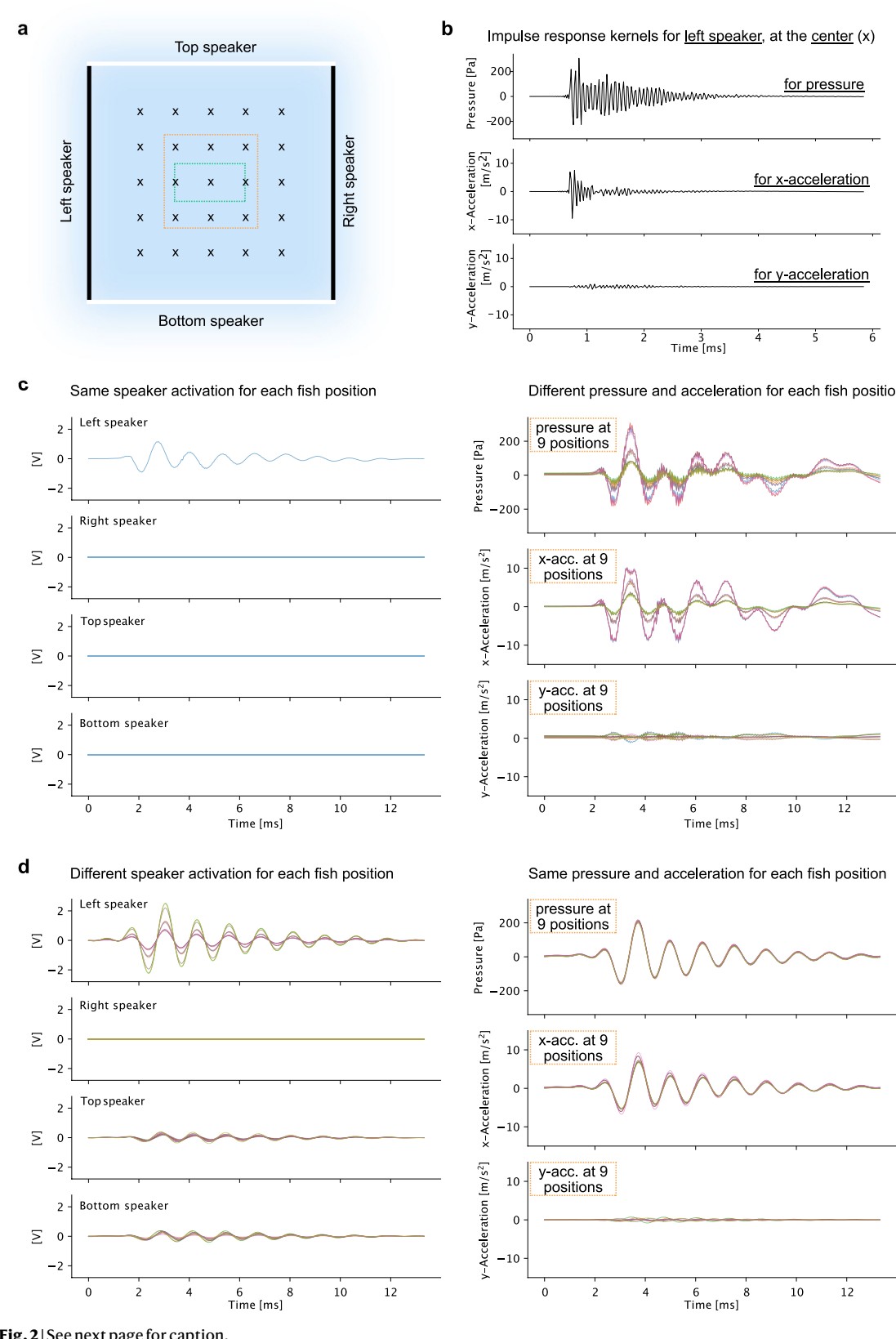

**Extended Data Fig. 2** | See next page for caption.

**Extended Data Fig. 2 | Sound targeting cancels reverb effects. a**, Schematic of sound calibration points (x) inside the inner tank. Green dotted rectangle: trigger zone. Orange dotted square: 9 measurements and target locations shown in **c-d. b**, Impulse response kernel for the left speaker at the center calibration point for three sound cues: pressure, $x$ acceleration, and $y$ acceleration. Pressure and $x/y$ acceleration impulse responses were recorded for each speaker and each position on the grid, totaling to 3 x 4 x 25 = 300 waveforms capturing the acoustic properties of the tank. **c**, Naive playback paradigm (not used): The same waveform is played back via the left speaker, and sound cues are recorded at nine positions. Left: constant speaker signals. Right: Pressure and $x/y$ acceleration measured at nine positions, spanning a 4.5 cm x 4.5 cm square at the center of the inner tank, vary considerably across this area, and the measured waveform does not resemble the playback waveform. **d**, Sound conditioning paradigm (used): Based on the impulse responses, different speaker signals are calculated for each target position. Left: speaker signals are different for each target position. Top and bottom speakers help to cancel $y$ acceleration. Right: Pressure and $x/y$ acceleration measured at the same nine positions. The waveforms resemble the defined target waveforms and are stable across positions.

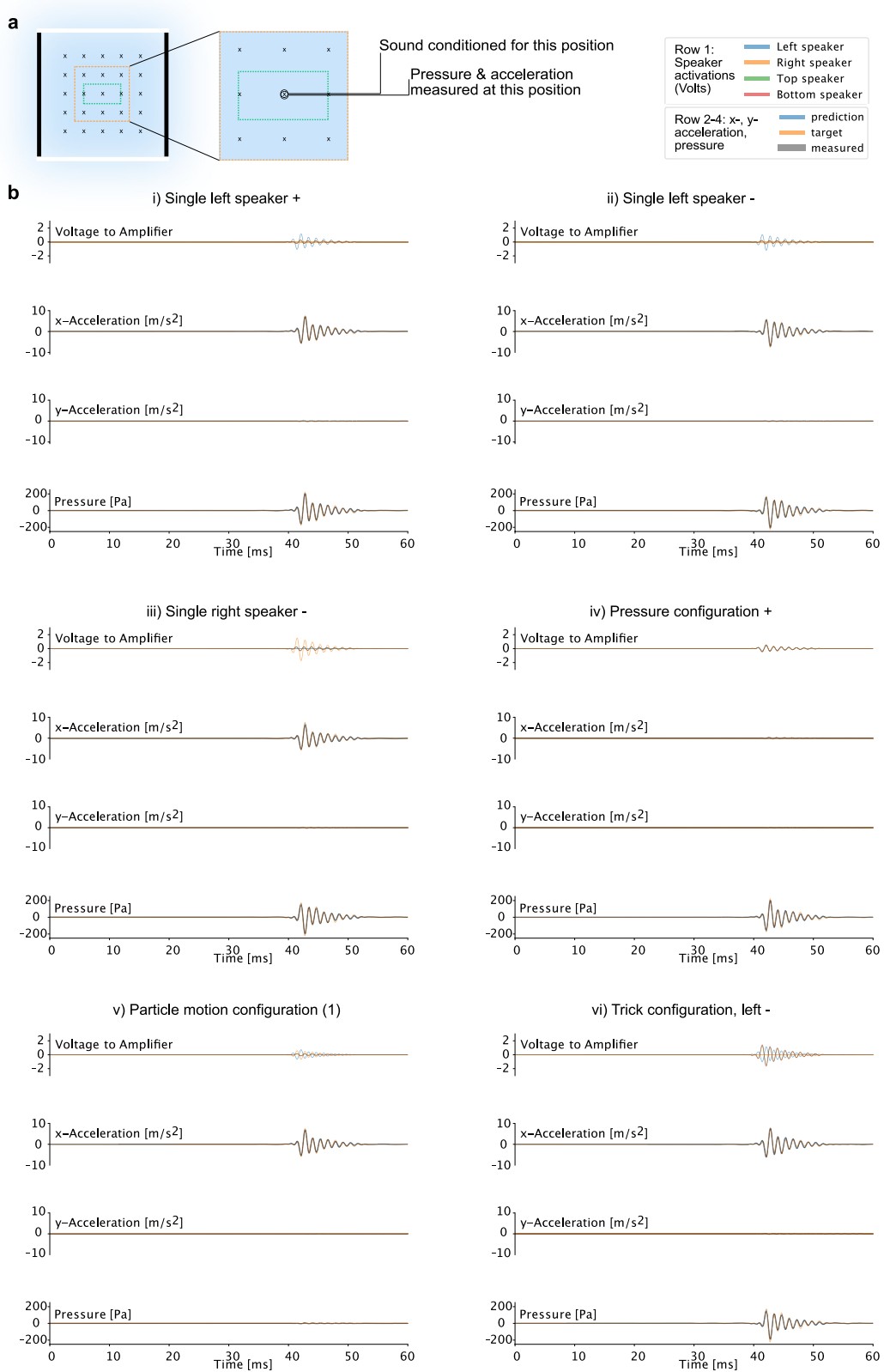

**Extended Data Fig. 3 | Accuracy of six exemplary sound stimuli. a**, Schematic of sound calibration points (x) inside the inner tank, indicating the sound target position and recording position for speaker signals and recordings shown in **b**. **b**, Pressure and acceleration measurements for sound configurations i) - vi) (see Fig. 2c for schematic), targeted for the inner tank's center calibration point and measured at the same position. Target waveforms (target) are defined as pressure, *x* accelerations, and *y* accelerations. Peak pressure: 223.87 Pa, peak *x* acceleration: 7.59 m/s². The speaker signals (top panels in i-vi) are calculated based on impulse responses to deliver the target waveforms to the target position with the constraint that, e.g., the right speaker is inactive in the left speaker configurations (i & ii) and in the left speaker trick configurations (vi). For a given speaker signal, convolution with the respective impulse response kernels predicts (prediction) the measured waveforms.

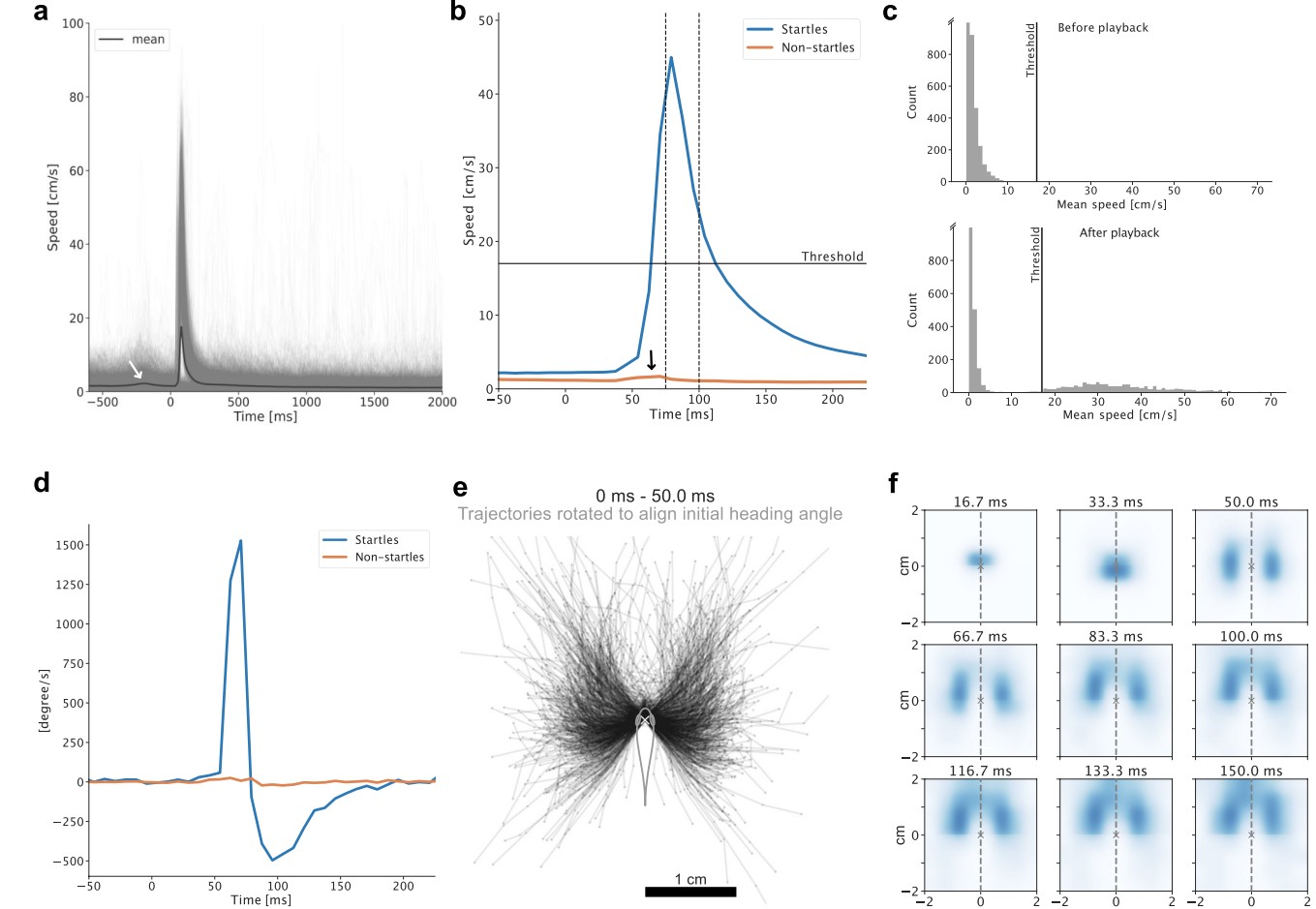

**Extended Data Fig. 4 | Startle detection and dynamics. a**, Fish translational speed, shown for all n = 3798 playbacks across twelve stimulus configurations, and across 65 fish aligned to playback trigger at t = 0 ms (thin lines). The mean speed across playbacks is shown as a thick line. To trigger a playback, the fish had to swim into the trigger zone. This explains the small increase in mean speed prior to t = 0 ms (white arrow). **b**, We used a 25 ms time window (between vertical dashed lines) and classified trials as startle trials if the temporal average speed across this time window exceeded a threshold of 17 cm/s (horizontal line). The remaining ones were classified as non-startle trials. The threshold was set such that the average speed across non-startle trials increased slightly after playback (black arrow). This reflects a conservative choice for startle detection, classifying a few startles as non-startles rather than classifying non-startles as startles. **c**, Top: The average speeds computed across a 25 ms time window before each playback are below the selection threshold of 17 cm/s. Bottom: The average speed distribution within the 25 ms time window indicated in **b**, after playback, is bimodal. Hence, startles can be readily identified by a speed threshold. **d**, Body bend: The average absolute angle taken between six edges along the fish's body axis after playback across startles increases sharply. It remains constant if averaged across non-startles. Startles are initiated by a strong body bend. **e**, Centered trajectories for all playbacks with startle reaction (n = 1415) across all 65 fish after aligning initial fish heading. The first 50 ms of startle responses are shown. **f**, Normalized heatmaps of the fish head position at consecutive times relative to the position at the start of the startle response. The top row shows snapshots of the trajectories shown in **e**. After a fast initial displacement to the left or right, *D. cerebrum* continues on a forward trajectory.

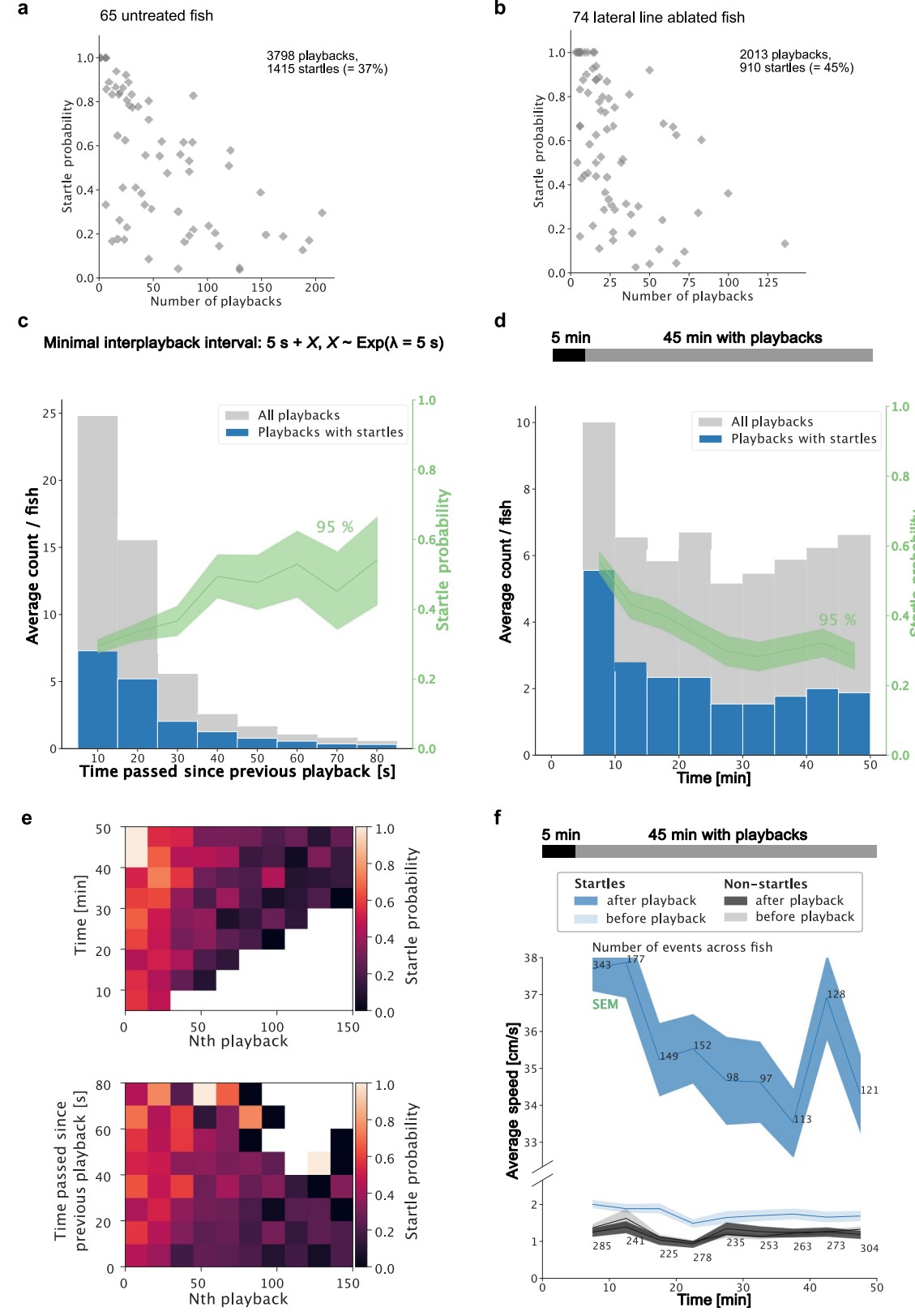

**Extended Data Fig. 5** | See next page for caption.

**Extended Data Fig. 5 | Habituation of the startle response. a-b**, Startle probability of each fish across all sound configurations as a function of the number of playbacks triggered by the fish. **a**, For main experiment (in untreated fish). **b**, For experiment in neomycin-treated fish. **c-f**, Details on habituation, data from main experiment (in untreated fish). **c**, Startle probability (the fraction of playbacks with startles, computed per bin) increases with time since the previous playback (green curve, the 95% confidence interval is given by Wilson scores). **d**, Startle probability (the fraction of playbacks with startles, computed per bin) decreases throughout the recording (green curve, the 95% confidence interval is given by Wilson scores). **e**, Startle probability as 2D binned statistics to show the interaction between time in recording and time since the last playback with the number of previous playbacks (Nth playback); white: no data. **f**, Average speeds over the course of the recording. Because fish that startle after playback have a higher speed before playback (see also Extended Data Fig. 4b), a decreased swimming speed could explain the decreasing startle probability over the course of the recording shown in **d**. Average speed is computed across 25 ms at a time 1 s before sound onset (before playback) and across 25 ms after sound onset (after playback), separately for startle and non-startle trials. The confidence interval indicates the standard error of the mean. Note the split y-axis. The speed of fish before playback stayed constant over the course of the recording. Hence habituation is not explained by a decrease in swimming activity over 50 min. **a-f**, Together, startle probability increases if the fish is in a fast swimming state and decreases, the less time has passed since the last playback and the more playbacks have been played previously (both signs of habituation).

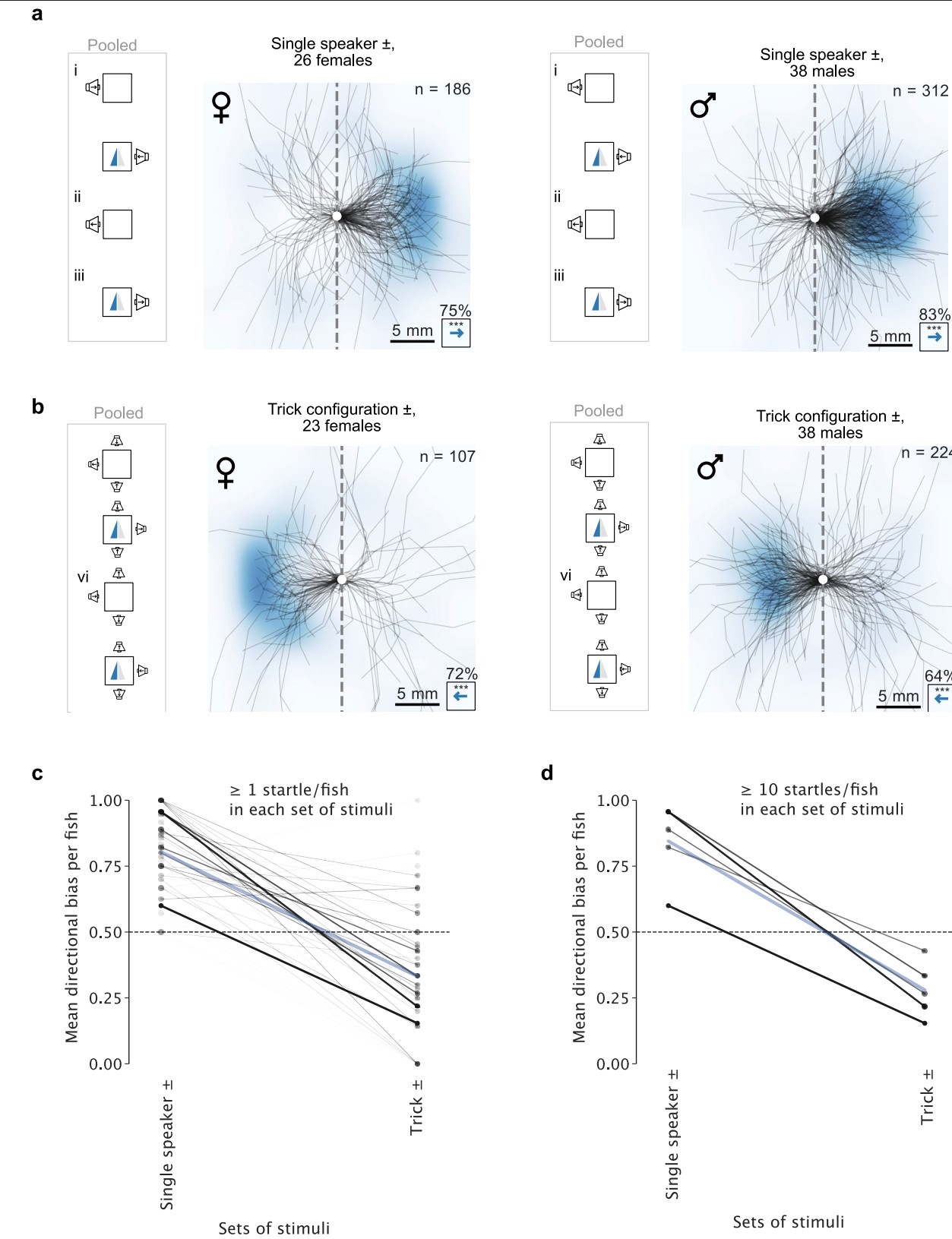

**Extended Data Fig. 6** | See next page for caption.

**Extended Data Fig. 6 | Directional startles are present for both sexes and in individual fish.** Centered startle trajectories for single speaker configurations and trick configurations, same data as shown in Fig. 2f but separated by sex. Both male and female *D. cerebrum* startle away from single active speakers and startle towards the active speaker in the trick configuration. **a**, Single speaker configuration. Left: females (75% of 186 startles away from the speaker in 26 female fish that startled, two-sided binomial test: $p = 9 \times 10^{-12}$). Right: males (83% of 312 startles away from the speaker in 38 male fish that startled, two-sided binomial test: $p = 1 \times 10^{-33}$). **b**, Trick configuration. Left: females (72% of 107 startles towards the speaker in 23 female fish that startled, two-sided binomial test: $p = 6 \times 10^{-6}$). Right: males (64% of 224 startles towards the speaker in 38 fish that startled, two-sided binomial test: $p = 2 \times 10^{-5}$). **a-b**, Percentages indicate the fraction of startle displacements into the direction of the blue arrow. The heatmaps are normalized and smoothed 2D histograms over endpoint positions of the shown trajectories. **c**, To estimate the directional bias in individual fish, we selected fish with at least one startle in both the single speaker condition and the trick condition (N = 46) to compute the number of fish with a mean directional bias away from the speaker (bias > 0.5): 41 of 46 fish in the single speaker configuration, two-sided binomial test: $p = 4 \times 10^{-8}$, and 10 of 46 in the trick configuration, two-sided binomial test: $p = 2 \times 10^{-4}$. Hence, individual fish escaped away from the single left–right speaker but approached it in the trick condition. **d**, The mean directional bias computed across fish with ≥10 startles in both conditions (N = 5, a subset of **a**) amounted to a directional bias of mean ± std. = 84% ± 13% away from the speaker in the single speaker configuration and to 72% ± 9% towards the speaker in the trick configuration. **c-d**, Line thickness is proportional to the number of minimum startles in response to one of the two sets of stimuli. Blue line: average directional bias across fish directional biases.

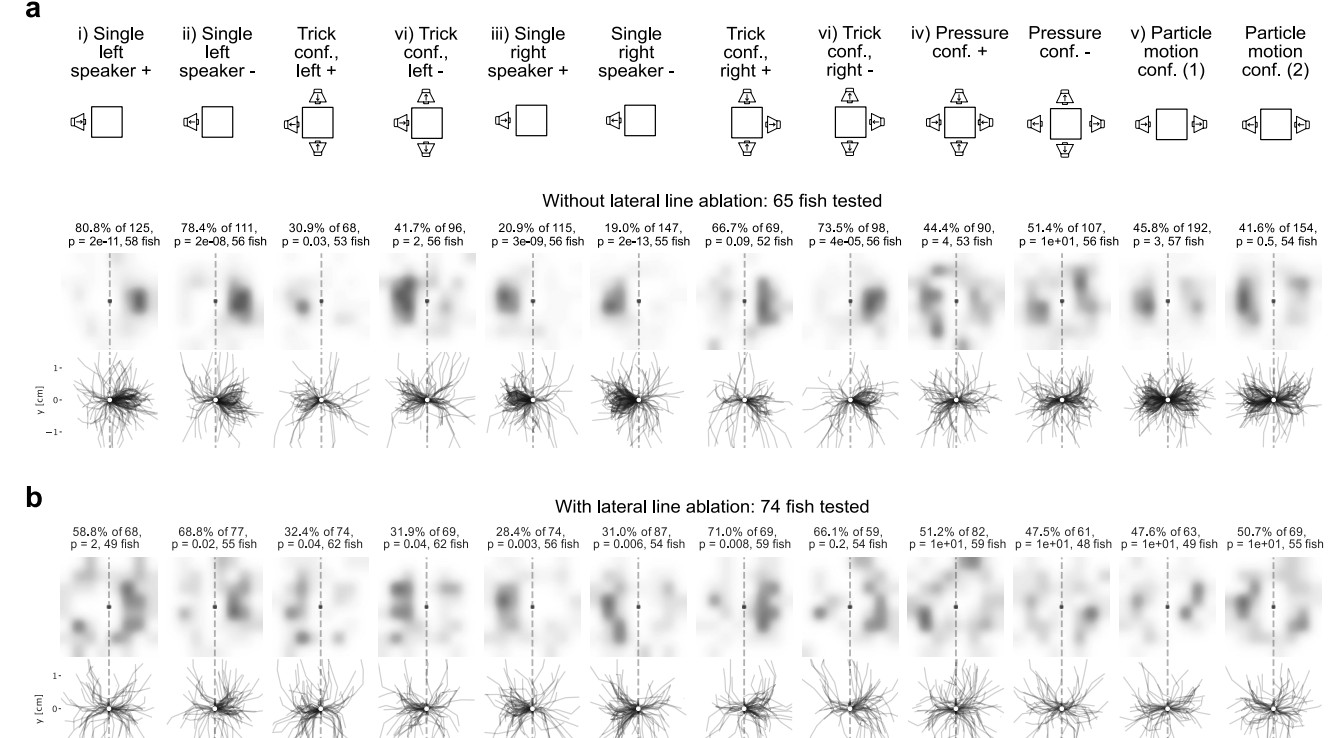

**Extended Data Fig. 7 | Data summary on all twelve stimuli for both cohorts.** Centered startle trajectories and displacement heatmaps for each stimulus used in the experiments, startle trials are pooled across all tested fish. **a**, for fish without an ablated lateral line. **b**, for fish with an ablated lateral line. **a-b**, Percentages indicate fraction of startles to the right. P-values report the probability that startles are unbiased in any direction (Two-sided binomial test, null hypothesis: p = 0.5, Bonferroni-corrected: n = 12 stimuli).

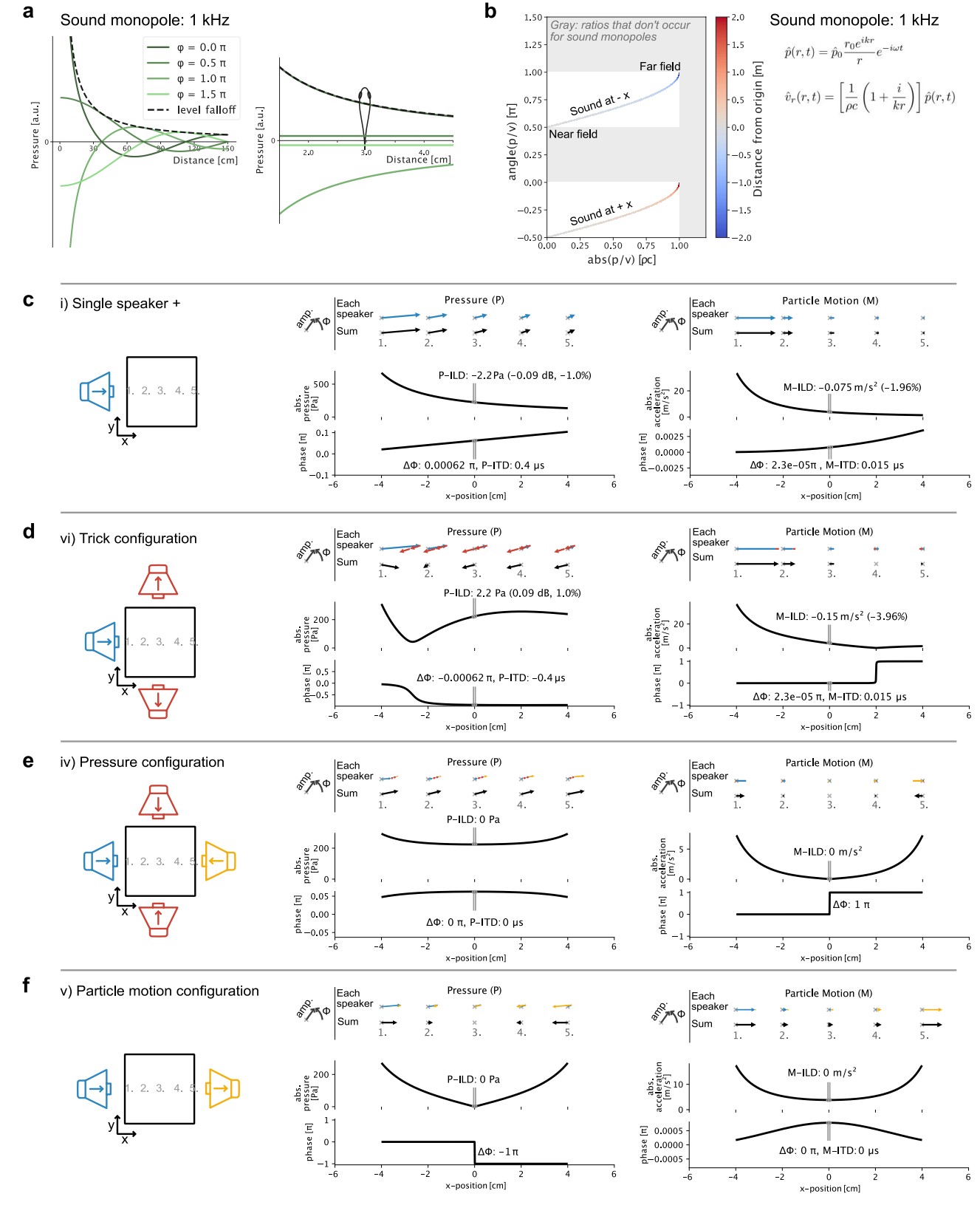

**Extended Data Fig. 8** | See next page for caption.

**Extended Data Fig. 8 | Sound monopoles and sound configurations. a**, Pressure level (dashed line) falloff next to a sound monopole at several phase snapshots of a propagating wave. Left: Falloff of a 1 kHz wave over 1.5 m. Right: The falloff across the width of the fish at 3 cm distance to a monopole sound source stems from the level falloff with distance. **b**, Both amplitude ratio and relative phase between pressure and particle velocity change in a distance-dependent manner. Both sound directions (-x, x) stay separate in the relative phase between pressure and particle velocity. **c-f**, Results of a simple model used to illustrate level and phase of pressure and motion along the horizontal x-axis of the inner tank. The idealized speaker activations in the different sound configurations are modeled as sinusoidal monopole sound sources (see pressure equation in **b**) located 6 cm away from the origin with frequency f = 780 Hz and speed of sound c = 1500 m/s. Acceleration is calculated from the spatial pressure gradient along the x-axis. The top rows are phasor representations of pressure or motion at five positions along the horizontal axis as indicated in the left cartoon. ILDs and ITDs are computed across a distance of 600 μm centered at the origin. See also Fig. 4 for a summary on ILD and ITD across sound configurations. **c**, In the single speaker configuration, P-ILD, P-ITD, M-ILD, and M-ITD could be interpreted as rightward cues by the fish. M-ITD is even smaller than P-ITD as motion phase propagates slower than pressure phase in the near field. **d**, In the trick configuration, P-ILD and P-ITD are inverted, while M-ILD and M-ITD remain unchanged as compared to the single speaker configuration. **c-d**, See Methods section on interaural cues for comparable P-ILD and M-ILD measurements in our setup. Note that we model monopoles in open water here, but the actual speakers are extended pressure sources in a tank. **e-f**, In both the pressure configuration and the particle motion configuration P-ILD, P-ITD, M-ILD, and M-ITD are zero or undefined.

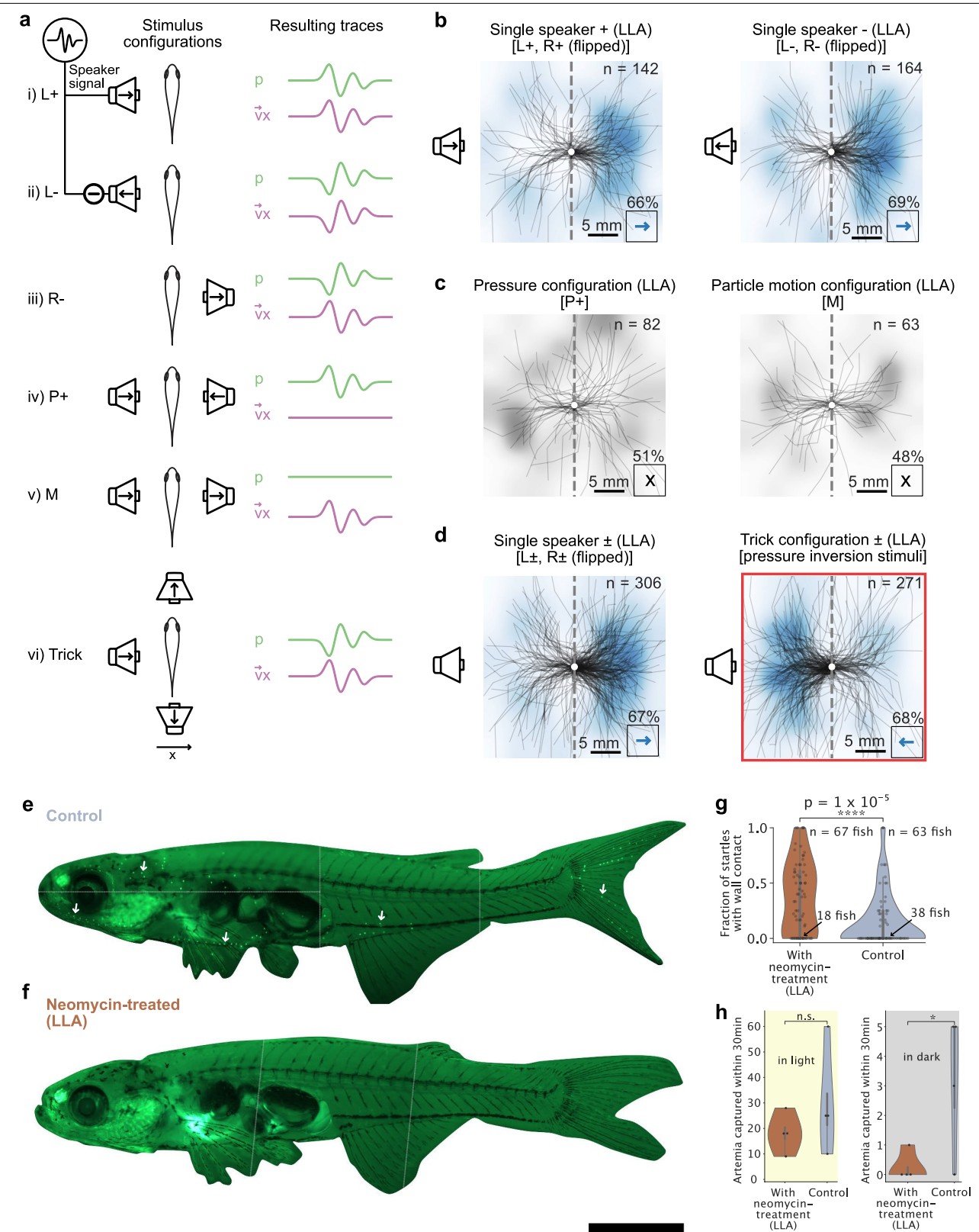

**Extended Data Fig. 9** | See next page for caption.

**Extended Data Fig. 9 | Directional behaviour is present after lateral line ablation. a-d,** Same as in Fig. 2c–f but for lateral line ablated (LLA) *D. cerebrum* **a,** Illustration of six different stimulus configurations. The cartoons of resulting traces illustrate the pressure and particle velocity relationship, idealized for a plane wave scenario. **b-d,** Centered startle trajectories after sound playback. **b,** Directional escapes away from a single speaker for a positive polarity waveform (66% escapes, two-sided binomial test: p = 0.0003 across n = 142 startles in 67 fish with startle trials in this sound configuration) and negative polarity waveform (69% escapes, two-sided binomial test: p = $1 \times 10^{-6}$ across n = 164 startles in 69 fish with startle trials in this sound configuration). **c,** Absence of directional bias in the positive polarity condition with only pressure (51%, two-sided binomial test: n.s.; p > 0.05) and only particle motion (48%, two-sided binomial test: n.s.; p > 0.05). See Extended Data Fig. 8 for opposite polarity results. **d,** Single speaker playbacks pooled over both polarities evoke a directional escape (67% escapes, two-sided binomial test: p = $1 \times 10^{-9}$ across n = 306 startles in 72 fish with startle trials in this sound configuration). Selective inversion of pressure polarity by an additional opposing pair of speakers along the orthogonal axis inverts the relative polarity between pressure and particle velocity. This implements the trick condition, in which the fish is tricked into performing startles approaching the active speaker (68% approaches, two-sided binomial test: p = $2 \times 10^{-9}$ across n = 271 startles in 73 fish with startle trials in this sound configuration). **b-d,** Percentages indicate the fraction of startle displacements into the direction of the blue arrow. The heatmaps are normalized and smoothed 2D histograms over endpoint positions of the shown trajectories (gray for single stimulus data, blue for pooled data). **e-f,** *D. cerebrum* stained with DASPEI, imaged under an epifluorescence microscope. Images are constructed from several fields of view (gray dotted lines). Scale bar: 2 mm **e,** In untreated fish, neuromasts are visible as green dots (see exemplary white arrows). **f,** After neomycin treatment, no neuromasts are visible, confirming lateral line ablation. **g-h,** Functional indicators of lateral line ablation. **g,** Neomycin-treated fish have significantly more wall contacts within the secondary escape trajectory, i.e. within ~500 ms (independent one-sided t-test p = $1 \times 10^{-5}$, after blind manual classification of the first ten startles in each recording into startles with and without wall contact. Some fish had less than ten startles, hence black dots are not always multiples of 10%). **h,** Left: In the light, untreated control shoals of mixed-sex adult *D. cerebrum* did not capture significantly more Artemia than neomycin-treated shoals (n = 4 shoals in each condition, independent one-sided t-test, p = 0.17). Right: In the dark, untreated shoals captured more Artemia (p = 0.024, independent one-sided t-test, n = 4 shoals in each condition), possibly using their intact lateral line sense. **g-h,** vertical gray stripe on violin plot indicates quartiles of data.

## a  Line scanning confocal reflectance microscope

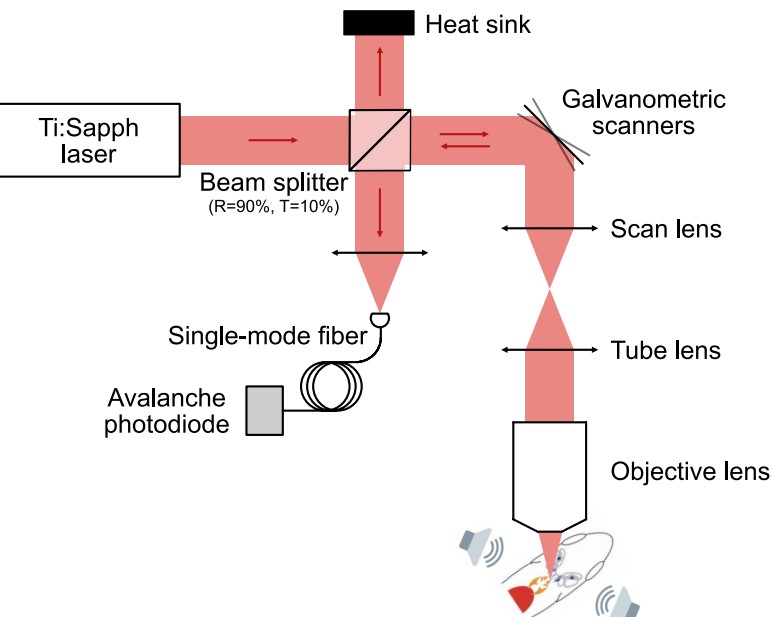

## b  Extracting motion amplitude and phase from line scan imaging

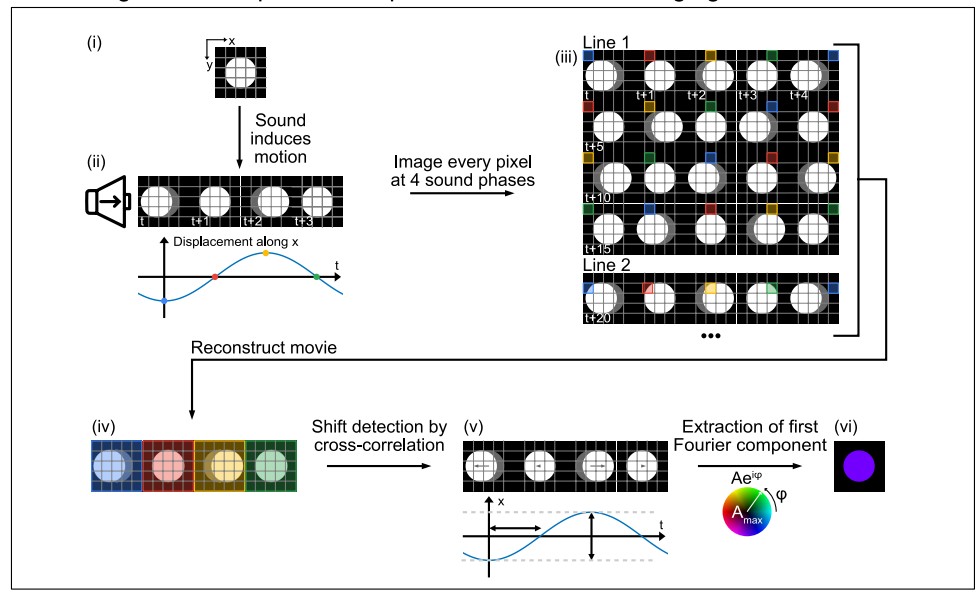

## c  Correcting motion phase for sound propagation across field-of-view

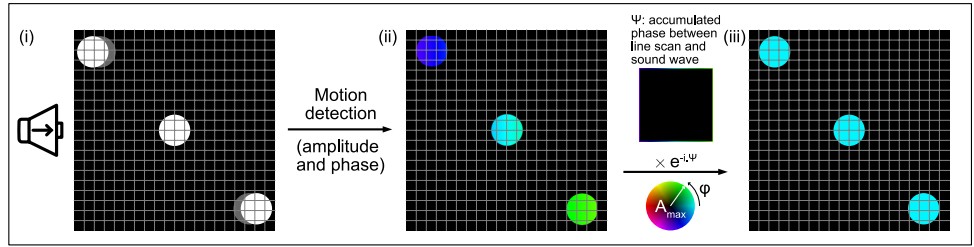

**Extended Data Fig. 10** | See next page for caption.

**Extended Data Fig. 10 | Method for extracting 2D phase maps of tissue motion with laser scanning confocal reflectance microscopy. a**, Experimental setup (see Methods section on vibrometry for details on the acoustic stimulation system). **b**, Illustration of phase map extraction: (i) A single bead is located within the field-of-view and (ii) oscillates in the horizontal x direction during acoustic stimulation. (iii) The bead is imaged with a laser scanning microscope, with each pixel being acquired at a different time. Line-scan and acoustic stimulation are synchronized to probe the bead at four different phases (blue: 0, red: π/2, yellow: π, green: 3π/2). (iv) Data are reshaped to reconstruct the full movie of the bead motion. (v) The bead displacement is computed using a cross-correlation-based algorithm (see Methods). (vi) The amplitude and phase of the first Fourier component of the bead displacement are extracted and plotted respectively with hue and color. **c**, (i) The sound phase and consequently the bead displacement phase is drifting when the pressure wave propagates along the horizontal x direction. Different motion phases are then detected for objects at different locations in the field-of-view, although being stimulated by the same sound wave. (ii) This additional phase Ψ is subtracted to yield a phase map with free objects exhibiting the same phase for the same sound stimulation. The final phase relationship between various objects therefore only depends on the mechanical properties of the imaged structure.

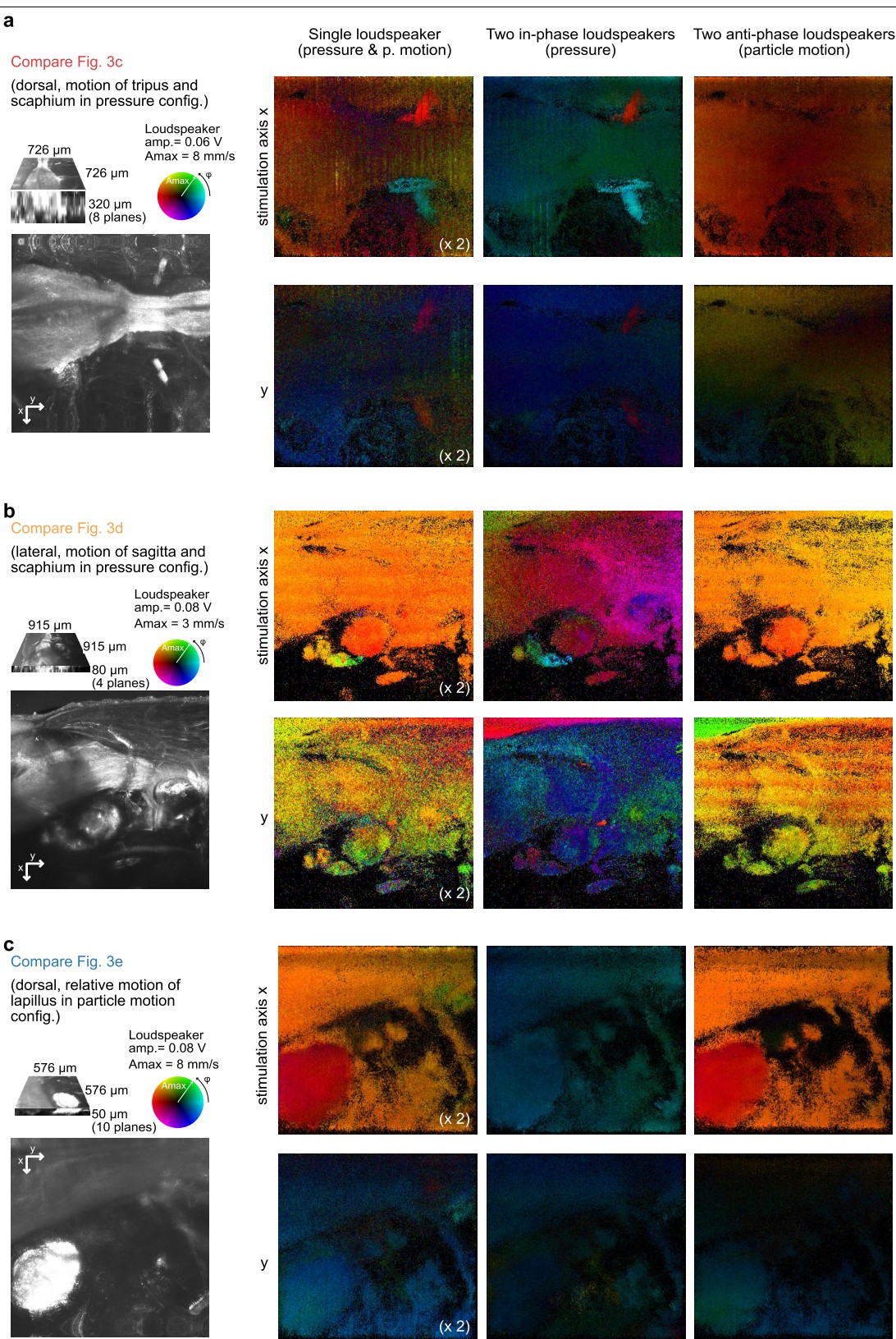

**a** Compare Fig. 3c
(dorsal, motion of tripus and scaphium in pressure config.)

726 µm
726 µm
320 µm (8 planes)

Loudspeaker amp.= 0.06 V
Amax = 8 mm/s

**b** Compare Fig. 3d
(lateral, motion of sagitta and scaphium in pressure config.)

915 µm
915 µm
80 µm (4 planes)

Loudspeaker amp.= 0.08 V
Amax = 3 mm/s

**c** Compare Fig. 3e
(dorsal, relative motion of lapillus in particle motion config.)

576 µm
576 µm
50 µm (10 planes)

Loudspeaker amp.= 0.08 V
Amax = 8 mm/s

Single loudspeaker (pressure & p. motion)
Two in-phase loudspeakers (pressure)
Two anti-phase loudspeakers (particle motion)

stimulation axis x

**Extended Data Fig. 11** | See next page for caption.

**Extended Data Fig. 11 | 2D optical vibrometry in three sound configurations.** Phase-amplitude particle velocity maps for 1 kHz component in response to 1 kHz speaker signals in the single speaker configuration (column 1), the pressure configuration (column 2), and the particle motion configuration (column 3). Depicted velocity amplitudes were scaled up by a factor of two in the single speaker condition to more readily compare responses to the pressure and motion configurations, where two speakers created a signal with approximately twice the pressure and motion amplitude. Across all three panels **a-c**, single-speaker maps resemble the addition of the pressure and motion maps. Sound stimulation was along the x-axis in all cases. See Fig. 3a for the location of depicted views and Fig. 3c–e for names of anatomical regions. **a**, Tripus tip and scaphium contract along the fish's medial-lateral axis (x) in the pressure configuration but not in the particle motion configuration. **b**, The scaphium and the sagitta move at different phases along the fish's ventral-dorsal axis (x) relative to surrounding tissue in the pressure configuration but not in the particle motion configuration. **c**, The lapillus moves with a phase lag to surrounding tissue along the medial-lateral axis (x) in the particle motion configuration but not the pressure configuration.

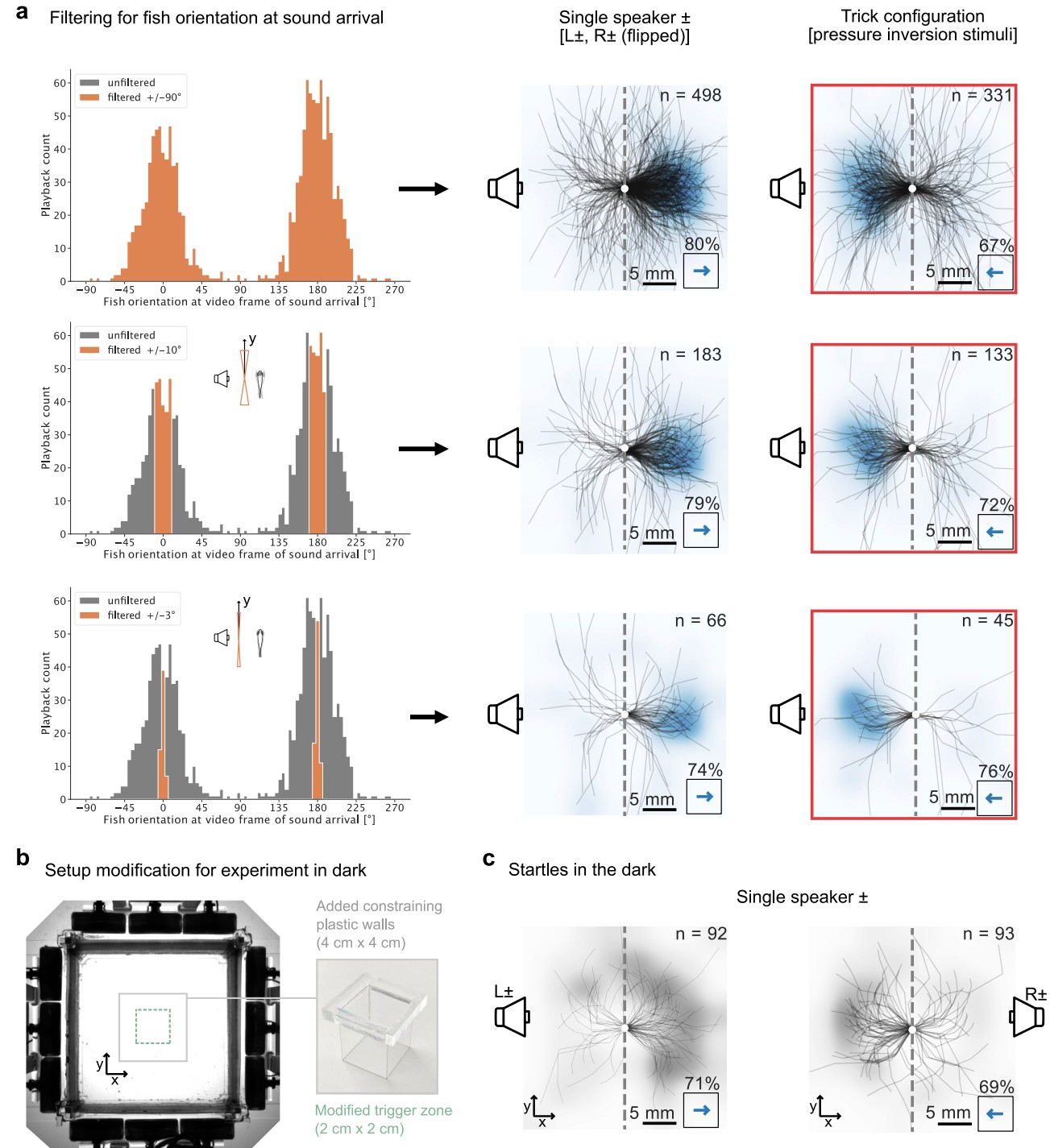

**Extended Data Fig. 12 | Excluding rostro-caudal swim bladder mechanism and cryptic visual cues. a**, To test the hypothesis that the pair of rostral and caudal swim bladder could implement a P-ILD or P-ITD sensor along the rostrocaudal axis, we filtered for playbacks where *D. cerebrum* was aligned orthogonal to the left–right axis during sound arrival at the fish location. Top: all data, same as Fig. 2f. Middle: filtered for ±10° alignment to y-axis, Single speaker (79% escapes, two-sided binomial test: p = 6 × 10⁻¹⁶, n = 183 startle trials in N = 59 fish), Trick configuration (72% escapes, two-sided binomial test: p = 3 × 10⁻⁷, n = 133 startle trials in N = 56 fish). Bottom: filtered for ± 3° alignment to y-axis, Single speaker (74% escapes, two-sided binomial test: p = 1 × 10⁻⁴, n = 66 startle trials in N = 47 fish), Trick configuration (76% escapes, two-sided binomial test: p = 8 × 10⁻⁴, n = 45 startle trials in N = 45 fish). **b-c**, Replication of the experiment in the dark to exclude the possibility of unknown visual cues. **b**, Top view of the modified setup for playback experiment in the dark. In the dark, the white inner tank walls could no longer prompt *D. cerebrum* to swim orthogonal to the left–right axis. To nevertheless trigger playbacks at the center of the tank with the fish being aligned orthogonally within a 45° cone, we added an additional constraining tank made from thin transparent plastic, thereby increasing the likelihood of triggering playback. **c**, Centered startle trajectory for experiment in the dark. Left: activation of the left speaker leads to rightward startles (71%, two-sided binomial test: p = 9 × 10⁻⁵, n = 92 startle trials in N = 43 fish). Right: activation of the right speaker leads to leftward startles (69%, two-sided binomial test: p = 0.0004, n = 93 startle trials in N = 43 fish).

# Reporting Summary

## Statistics

For all statistical analyses, confirm that the following items are present in the figure legend, table legend, main text, or Methods section.

| n/a | Confirmed | |
|---|---|---|
| ☐ | ☒ | The exact sample size (*n*) for each experimental group/condition, given as a discrete number and unit of measurement |
| ☒ | ☐ | A statement on whether measurements were taken from distinct samples or whether the same sample was measured repeatedly |
| ☐ | ☒ | The statistical test(s) used AND whether they are one- or two-sided<br>*Only common tests should be described solely by name; describe more complex techniques in the Methods section.* |
| ☐ | ☒ | A description of all covariates tested |
| ☐ | ☒ | A description of any assumptions or corrections, such as tests of normality and adjustment for multiple comparisons |
| ☐ | ☒ | A full description of the statistical parameters including central tendency (e.g. means) or other basic estimates (e.g. regression coefficient) AND variation (e.g. standard deviation) or associated estimates of uncertainty (e.g. confidence intervals) |
| ☐ | ☒ | For null hypothesis testing, the test statistic (e.g. *F*, *t*, *r*) with confidence intervals, effect sizes, degrees of freedom and *P* value noted<br>*Give P values as exact values whenever suitable.* |
| ☒ | ☐ | For Bayesian analysis, information on the choice of priors and Markov chain Monte Carlo settings |
| ☒ | ☐ | For hierarchical and complex designs, identification of the appropriate level for tests and full reporting of outcomes |
| ☒ | ☐ | Estimates of effect sizes (e.g. Cohen's *d*, Pearson's *r*), indicating how they were calculated |

*Our web collection on statistics for biologists contains articles on many of the points above.*

## Software and code

Policy information about availability of computer code

| Data collection | We used custom software written in Python 3 and MATLAB (v2019b) (https://github.com/danionella/veith_et_al_2024) as well as LSMAQ for laser scanning microscopy (https://github.com/danionella/lsmaq) |
|---|---|
| Data analysis | We used custom software written in Python 3 and MATLAB (v2019b), as well as SLEAP (v1.3), BIOMEDISA (v23), FIJI (v1.5), 3D Slicer (v5.6) |

For manuscripts utilizing custom algorithms or software that are central to the research but not yet described in published literature, software must be made available to editors and reviewers. We strongly encourage code deposition in a community repository (e.g. GitHub). See the Nature Portfolio guidelines for submitting code & software for further information.

## Data

Policy information about availability of data

All manuscripts must include a data availability statement. This statement should provide the following information, where applicable:
- Accession codes, unique identifiers, or web links for publicly available datasets
- A description of any restrictions on data availability
- For clinical datasets or third party data, please ensure that the statement adheres to our policy

Fish trajectory, micro-CT, and vibrometry data have been deposited in the G-Node repository at: gin.g-node.org/danionella/veith_et_al_2024

# Research involving human participants, their data, or biological material

Policy information about studies with human participants or human data. See also policy information about sex, gender (identity/presentation), and sexual orientation and race, ethnicity and racism.

| | |
|---|---|
| Reporting on sex and gender | N/A |
| Reporting on race, ethnicity, or other socially relevant groupings | N/A |
| Population characteristics | N/A |
| Recruitment | N/A |
| Ethics oversight | N/A |

Note that full information on the approval of the study protocol must also be provided in the manuscript.

# Field-specific reporting

Please select the one below that is the best fit for your research. If you are not sure, read the appropriate sections before making your selection.

☒ Life sciences    ☐ Behavioural & social sciences    ☐ Ecological, evolutionary & environmental sciences

For a reference copy of the document with all sections, see nature.com/documents/nr-reporting-summary-flat.pdf

# Life sciences study design

All studies must disclose on these points even when the disclosure is negative.

| | |
|---|---|
| Sample size | The sample size was chosen based on the standards in the field and estimated behavioral variability. |
| Data exclusions | No data were excluded. |
| Replication | We successfully replicated our findings in males (39), females (26), and lateral-line ablated fish (74). Observations shown in Figure 3 c-e were repeated once, four times, and once in other fish, respectively, with similar results. |
| Randomization | There was no group allocation which needed to be randomized. |
| Blinding | There was no group allocation which would have required blinding. |

# Reporting for specific materials, systems and methods

We require information from authors about some types of materials, experimental systems and methods used in many studies. Here, indicate whether each material, system or method listed is relevant to your study. If you are not sure if a list item applies to your research, read the appropriate section before selecting a response.

## Materials & experimental systems

| n/a | Involved in the study |
|---|---|
| ☒ ☐ | Antibodies |
| ☒ ☐ | Eukaryotic cell lines |
| ☒ ☐ | Palaeontology and archaeology |
| ☐ ☒ | Animals and other organisms |
| ☒ ☐ | Clinical data |
| ☒ ☐ | Dual use research of concern |
| ☒ ☐ | Plants |

## Methods

| n/a | Involved in the study |
|---|---|
| ☒ ☐ | ChIP-seq |
| ☒ ☐ | Flow cytometry |
| ☒ ☐ | MRI-based neuroimaging |

## Animals and other research organisms

Policy information about studies involving animals; ARRIVE guidelines recommended for reporting animal research, and Sex and Gender in Research

| | |
|---|---|
| Laboratory animals | Wildtype adult (≥ 4 month old) Danionella cerebrum |
| Wild animals | No wild animals were used in this study |
| Reporting on sex | Findings apply to both sexes and sexes have been indicated explicitly |
| Field-collected samples | No field-collected samples were used in this study |
| Ethics oversight | All animal experiments conformed to Berlin state, German federal and European Union animal welfare regulations and were approved by the LAGeSo, the Berlin authority for animal experiments. |

Note that full information on the approval of the study protocol must also be provided in the manuscript.

## Plants

| | |
|---|---|
| Seed stocks | *Report on the source of all seed stocks or other plant material used. If applicable, state the seed stock centre and catalogue number. If plant specimens were collected from the field, describe the collection location, date and sampling procedures.* |
| Novel plant genotypes | *Describe the methods by which all novel plant genotypes were produced. This includes those generated by transgenic approaches, gene editing, chemical/radiation-based mutagenesis and hybridization. For transgenic lines, describe the transformation method, the number of independent lines analyzed and the generation upon which experiments were performed. For gene-edited lines, describe the editor used, the endogenous sequence targeted for editing, the targeting guide RNA sequence (if applicable) and how the editor was applied.* |
| Authentication | *Describe any authentication procedures for each seed stock used or novel genotype generated. Describe any experiments used to assess the effect of a mutation and, where applicable, how potential secondary effects (e.g. second site T-DNA insertions, mosiacism, off-target gene editing) were examined.* |

