## [Peer Review File · Nature]

Manuscript Title: The mechanism for directional hearing in fish

Reviewer Comments & Author Rebuttals

Reviewer Reports on the Initial Version:

Referees' comments:

Referee #1 (Remarks to the Author):

Review of The mechanism for directional hearing in fish by Veith et al. This paper provides a major step forward in understanding sound localization in fishes. Fish sensory hair cells are intrinsically directional, but their readout is 180° ambiguous. In 1975, Schuijf proposed that fish could deduce the direction of sound source if they could compare hair cell directionality (vector) and swimbladder pressure sensitivity (scalar). This theory has dominated the field of fish hearing, but been hard to test. The authors use the model teleost *Danio rerio*, which is transparent and uses sound for communication, to test the Schuijf theory. In addition to the wise choice of *Danio rerio*, their major advances lie in the clever use of stimuli to separate particle motion from pressure stimuli, and use of linescan confocal analysis to differentiate between pressure and particle motion activation of the inner ear.

I do have a couple of concerns. Most notable is the authors' prediction, from observing endorgan activation via laser scanning, that the utricle is the main site for particle motion sensing in the horizontal plane. Depending on the species, the literature supports leading roles for saccule, and/or lagenar and/or utricle in particle motion sensing. Fay (1984) *Science* 225, 951–954, used particle motion stimulation and recordings from all 3 endorgans in goldfish. He wrote "It is now clear that the otolithic ears of the goldfish (and other fish species) code the sound-pressure waveform (through input to the saccule from the swim bladder) and the particle-motion waveform (through the direct inertial route used in these experiments) through directionally sensitive receptors." Later Fay recordings from the vocalizing toadfish supported sensitivity to particle motion from the saccule. There are other studies supporting particle motion sensitivity from the lagena. It would be better if the authors cited some of these studies. Instead they write "...saccule may be the end organ that mostly senses the pressure component of sound, in agreement with findings in other fishes", with these cited studies being all from one group. I suggest a brief discussion of the different lines of evidence for particle and pressure sensitivity with respect to fish phylogeny.

Along the same lines, on p. 9 the authors write 'These results suggest that ... the utricle is the main site for particle motion sensing in the horizontal plane via the direct pathway (Fig. 3g).: They have actually only stimulated by particle motion in one direction, so cannot rule out whether the saccule would vibrate in response to motion stimulation along its long axis, for example. This could also explain the discrepancy between this and the earlier studies referred to above that show particle motion sensitivity of the sacculus.

Another concern relates to the quality and interpretation of the laser scans. This is a novel use of the

technique, and the authors are to be congratulated. Nevertheless, more explanation of Figure 2 would be helpful.

A few comments:

line 3-4 (and stated again in introduction, l. 19): ...makes interaural cues negligible... UW cues are smaller, but whether they are negligible remains to be determined- especially since one of the hypotheses stated four lines below is that fish could be able to deal with '...minute interaural differences... It may be ruled out for the small *Danio*, but is not completely impossible for larger fish.

Results, l. 3: 'abrupt' sounds strikes me as a very imprecise description – I suggest 'transient sounds' – and why the signal is made in that strange way described on p. 20 –instead of just synthesizing a transient eludes me:

'We observed that *Danio* c. startle when we drop a cylindrical piece of rubber into the water. We recorded the pressure waveform of this sound...

Discussion, l. 5: The most recent model is in the paper by Sisneros et al (2016) – that fish use the intensity vector (a time average of motion and pressure components) – maybe this model should be mentioned explicitly.

I can't find the citation in text for Zedler, D. G., Fay, R. R., Alderks, P. W., Shaub, K. S. & Sisneros, J. A. Sound source localization by the plainfin midshipman fish, *J Acoust Soc Am* 127, 10 (2010).

Referee #2 (Remarks to the Author):

In this manuscript Veith and colleagues answer a long standing question related to directional hearing in aquatic animals. What makes this work stand out is the quantitative rigor and attention to experimental detail with which an apparent paradox in animal sensory processing is addressed.

The significant advance in the current manuscript is threefold:

- 1) The authors go a long way in identifying the actual mechanism and physical process by which directional hearing under water at tiny interaural distances could be implemented.
- 2) They go above and beyond in carefully testing a whole range of mutually exclusive – but biologically plausible – hypotheses of how fish might implement the hardware necessary for directional hearing. The combined application of pressure phase and particle motion sensing (implemented by two independent sensors and pathways) is convincing and persuasive.
- 3) They provide the ultimate test by cleverly applying the two different stimulus components with independent control. This allows for an elegant test of their hypothesis. The ultimate – and most persuasive – experiment is the selective inversion of pressure polarity by an additional opposing pair of speakers along the longitudinal axis. The application of this unnatural stimulus inverts the relative polarity between pressure and particle velocity and allows for the final test and confirmation of the hypothesis by eliciting startles in the opposite direction where fish now approach the active speaker.

This is very elegant indeed!

Implementing all experimental aspect of this project is a technical and conceptual tour de force and a truly heroic effort. The manuscript is beautifully written, and the complex and intricate topic is explained clearly. The author have managed to make a potentially dry and mathematically involved matter readily accessible to a broader audience.

Overall, I find this work constitutes a significant breakthrough in a long standing question on directional hearing in teleost fish and in aquatic animals in general. It provides a final and definitive answer to a long standing and important puzzle, and it will undoubtedly have a lasting impact on the field. I firmly believe that this paper will become a 'classic', it will be of great interest to the community and I support publication strongly, provided my concerns below can be addressed.

Comments:

Major:

1) Currently the weakest aspect of the manuscript is the rejection of the P-ILD model. This is compatible with the behavioral data, but requires two separate pressure sensors – one closer to the speaker than the other. The authors' basis for rejecting this model is that they do not find two such pressure sensors in their vibrometry imaging experiments. While the imaging technology is novel and elegant, and the execution thorough and impressive, the main result still remains a negative finding. Just because you cannot find something doesn't mean it isn't there. As such the results presented in Figure 3 are certainly compatible with the hypothesis but they do not provide as conclusive a proof as the behavioral experiments, where especially the "trick" configuration is particularly elegant and convincing. I am not sure I fully and completely understand the logic of the various speaker configurations as shown in Figure 2c, but the way I see it, the vi) "Trick" configuration could be improved as follows: use the M-only version (Fig. 1c v) to deliver the particle motion cue, instead of a single speaker. , i.e. a pair of "horizontally" placed speakers instead of just a single one, as currently depicted in vi). Then a clean particle motion stimulus can be delivered with minimal pressure contamination, and the pair of "vertically" placed speakers that deliver the inverted pressure wave, would be unincumbered by the single speaker that currently provides interference of the pressure wave. This interference might also explain the somewhat poorer performance in the "trick" experiments (67% vs 80%)? Importantly, if this reasoning makes sense, then this suggested extended configuration also would solve the inversion of the pressure gradient issue; since then the pressure gradient would just be nose-to-tail and not left-to-right.

2) Success of the lateral line ablations is confirmed by an absence of staining. This is not ultimately convincing, since, again, this is a negative result. Is it possible to add an additional control that uses a functional metric? Is there a behavior that the fish can only execute with intact neuromasts? Or is it possible to record some other physiological response whose absence after ablation would provide a more convincing result?

minor concerns/suggestions:

can the behavioral experiments shown in Figure 1 be repeated in darkness? I realize that the short delay and the symmetry in the set-up make it unlikely that the fish use some "hack" based on cryptic

visual cues – but it would be a nice control.

The numbering of the Extended Data Figures do not appear in ranked order in the text.

Referee #3 (Remarks to the Author):

Veith et al provide a technical tour de force and comprehensive paper on sound localization in a tiny transparent species that allows visualization of the brain and otolith organs within an intact fish. The fish bioacoustics world is divided into workers with specializations in engineering, acoustics, auditory and muscle physiology, experimental anatomy and histology, behavior and ecology. Rarely are more than two of these fields represented in a single paper. The authors use expertise in all of these fields (save ecology) to attack a difficult question, namely how a fish localizes sound, which has been a major unsettled issue in fish acoustics. The question is complicated due to the high speed of sound in water, which reduces cues that would be available to terrestrial animals and humans. Among other accomplishments, the authors are able set up different test conditions to differentiate between previous hypotheses. They provide support for Schuijff's notion that otophysans (fishes with Weberian ossicles that connect the swimbladder to the ears) utilize a combination of particle velocity and acoustic pressure. If memory serves, Schuijff's theory, previously hypothetical, has now been supported by the study whereas other competing ideas failed experimentally. The authors also accomplish this work in fish with intact and disabled lateral lines. Lateral lines potentially provide directional auditory cues although they are primarily tuned to lower frequency water movement. In this case the ears are sufficient and the lateral line is not required for a directional escape response. Among other feats, the authors use laser vibrometry to demonstrate differences in how the otoliths respond individually to pressure or particle velocity. Such work would previously have required anesthetic and surgical exposure that could affect vibratory behavior of end organs. The paper is challenging due to the inclusion of a large amount of diverse information but is extremely well written, clear and does a good job of summarizing past literature.

I have a couple of minor suggestions.

Introduction, para 3. Delete plainfin or put it before midshipman.

Fig. 2. The authors provide an illustration of the stimulus waveforms. Given the acoustic complexity of small underwater tanks, I suspect the actual signals were not quite this pretty. I think a statement might be warranted. I also note that the results stand so the pressure and velocity signal were clearly sufficient to support the results in the paper.

Behavioral set up and protocol. Pg 16.

I was originally going to ask for a little more detail such as dimensions of the outer tank, but I note that much of this is given in the figure. I think it would be worthwhile to specify the distance between the speaker and the choice point, how the speaker was mounted, and the effect of the plastic inner tank on acoustic transmission. The plastic inner tank likely has an acoustic impedance similar to water and would not be a major problem. Still, it would be nice to state how signal amplitude and waveform were or were not affected. What does transparent ground of the inner tank mean? Fish column is not exactly clear.

Michael L Fine

Author Rebuttals to Initial Comments

We thank all reviewers for their very helpful suggestions and constructive critique. Reviewers' comments are in blue, our responses are in black.

Referee #1 (Remarks to the Author):

Review of The mechanism for directional hearing in fish by Veith et al. This paper provides a major step forward in understanding sound localization in fishes. Fish sensory hair cells are intrinsically directional, but their readout is 180° ambiguous. In 1975, Schuijf proposed that fish could deduce the direction of sound source if they could compare hair cell directionality (vector) and swimbladder pressure sensitivity (scalar). This theory has dominated the field of fish hearing, but been hard to test. The authors use the model teleost *Danionella*, which is transparent and uses sound for communication, to test the Schuijf theory. In addition to the wise choice of *Danionella*, their major advances lie in the clever use of stimuli to separate particle motion from pressure stimuli, and use of linescan confocal analysis to differentiate between pressure and particle motion activation of the inner ear.

I do have a couple of concerns.

Most notable is the authors' prediction, from observing endorgan activation via laser scanning, that the utricle is the main site for particle motion sensing in the horizontal plane. Depending on the species, the literature supports leading roles for saccule, and/or lagenar and/or utricle in particle motion sensing. Fay (1984) *Science* 225, 951-954, used particle motion stimulation and recordings from all 3 endorgans in goldfish. He wrote "It is now clear that the otolithic ears of the goldfish (and other fish species) code the sound-pressure waveform (through input to the saccule from the swim bladder) and the particle-motion waveform (through the direct inertial route used in these experiments) through directionally sensitive receptors." Later Fay recordings from the vocalizing toadfish supported sensitivity to particle motion from the saccule. There are other studies supporting particle motion sensitivity from the lagena. It would be better if the authors cited some of these studies. Instead they write "...saccule may be the end organ that mostly senses the pressure component of sound, in agreement with findings in other fishes", with these cited studies being all from one group. I suggest a brief discussion of the different lines of evidence for particle and pressure sensitivity with respect to fish phylogeny. Along the same lines, on p. 9 the authors write 'These results suggest that ... the utricle is the main site for particle motion sensing in the horizontal plane via the direct pathway (Fig. 3g).: They have actually only stimulated by particle motion in one direction, so cannot rule out whether the saccule would vibrate in response to motion stimulation along its long axis, for example. This could also explain the discrepancy between this and the earlier studies referred to above that show particle motion sensitivity of the sacculus.

Thank you for these very good points. Our use of “mostly” was indeed misleading. We meant to identify the saccule as the main end organ for pressure sensing, not to rule out that it may respond to direct particle motion. The updated sentence should make this clearer:

We concluded that an indirect pressure sensing pathway exists in *Danionella c.* and that the saccule may be **its main** the end organ ~~that mostly senses the pressure component of sound~~, in agreement with findings in other fishes [36,37,61,62].

We also updated the following paragraph:

[...] We did not detect such relative motion for the tripus, the sagitta and the asteriscus. **Note that we used stimuli with particle motion along the mediolateral axis relevant for the left/right startle behavior. To support directional hearing in three dimensions, *Danionella c.* may have further direct motion sensing pathways along additional axes, in line with particle motion tuning in saccular [63,64] or lagenar [63,65] afferents in other species.**

Another concern relates to the quality and interpretation of the laser scans. This is a novel use of the technique, and the authors are to be congratulated. Nevertheless, more explanation of Figure 2 would be helpful.

Following this suggestion, we added a new extended data figure to explain in more detail the laser-scanning-based vibrometry technique used in Figure 3. The corresponding methods section has been updated to point at the relevant panels of this figure at each step of the process. In short, by synchronizing the laser scan with sound playback, we can reconstruct a stroboscopic video of motion over time. With the help of cross-correlation between consecutive frames, we calculate x-y displacement vector for each pixel as a function of time. Taking the temporal Fourier transform of this displacement field results in the motion phase and amplitude maps shown in the paper. The new figure is pasted below:

a. Line scanning confocal reflectance microscope

b. Extracting motion amplitude and phase from line scan imaging

c. Correcting motion phase for sound propagation across field-of-view

Extended Data Fig 10 | Method for extracting 2D phase maps of tissue motion with laser scanning confocal reflectance microscopy: **a**, Experimental setup (see Methods section on vibrometry for details on the acoustic stimulation system). **b**, Phase map extraction: (i) A single bead is located within the field-of-view and (ii) oscillates in the horizontal x direction upon acoustic stimulation. (iii) The bead is imaged with a laser scanning microscope, with each pixel being acquired at a different time. Line-scan and acoustic stimulation are synchronized to probe the bead at four different phases (blue: 0, red: $\pi/2$, yellow: π , green: $3\pi/2$). (iv) Data are reshaped to reconstruct the full movie of the bead motion. (v) The bead displacement is computed using a cross-correlation-based algorithm, implemented in the PIVlab software [80]. (vi) The amplitude and phase of the first Fourier component of the bead displacement are extracted and plotted respectively with hue and color. **c**, (i) The sound phase and consequently the bead displacement phase is drifting when the pressure wave propagates along the horizontal x direction. Different motion phases are then detected for objects at different locations in the field-of-view, although being stimulated by the same sound wave. (ii) This additional phase Ψ is subtracted to yield a phase map with free objects exhibiting the same phase for the same sound stimulation. The final phase relationship between various objects therefore only depends on the mechanical properties of the imaged structure.

A few comments:

line 3-4 (and stated again in introduction, l. 19): ...makes interaural cues negligible... UW cues are smaller, but whether they are negligible remains to be determined- especially since one of the hypotheses stated four lines below is that fish could be able to deal with '...minute interaural differences... It may be ruled out for the small *Danionella*, but is not completely impossible for larger fish.

We agree, the wording should apply to all fish, not just *Danionella*. We replaced “negligible” with “very small” in both places.

Results, l. 3: ‘abrupt’ sounds strikes me as a very imprecise description - I suggest ‘transient sounds’ - and why the signal is made in that strange way described on p. 20 -instead of just synthesizing a transient eludes me:

‘We observed that *Danionella c.* startle when we drop a cylindrical piece of rubber into the water. We recorded the pressure waveform of this sound...

Thank you for this suggestion to clarify wording. We replaced “abrupt” with “transient”, as suggested. We chose this stimulus after noticing that it caused a strong startle response and then stuck with what worked well.

Discussion, l. 5: The most recent model is in the paper by Sisneros et al (2016) - that fish use the intensity vector (a time average of motion and pressure components) - maybe this model should be mentioned explicitly.

We added a new paragraph to the discussion to describe the model of Sisneros and Rogers in more detail:

“Sisneros and Rogers have recently extended Schuijff’s model by proposing that fish compute the time-averaged product of pressure and motion (the acoustic intensity vector) (Sisneros & Rogers 2016). This theory can account for phonotaxis of plainfin midshipmen towards monopole and dipole sources (Zeddies 2010, Zeddies 2012). Future neurophysiological work may test whether *Danionella c.* implements Schuijff’s model this way.”

I can’t find the citation in text for Zeddies, D. G., Fay, R. R., Alderks, P. W., Shaub, K. S. & Sisneros, J. A. Sound source localization by the plainfin midshipman fish, . J Acoust Soc Am 127, 10 (2010).

This reference is cited in the third paragraph of the introduction (after “midshipman”). It is now also cited in the updated discussion section quoted above.

Referee #2 (Remarks to the Author):

In this manuscript Veith and colleagues answer a long standing question related to directional hearing in aquatic animals. What makes this work stand out is the quantitative rigor and attention to experimental detail with which an apparent paradox in animal sensory processing is addressed.

The significant advance in the current manuscript is threefold:

- 1) The authors go a long way in identifying the actual mechanism and physical process by which directional hearing under water at tiny interaural distances could be implemented.
- 2) They go above and beyond in carefully testing a whole range of mutually exclusive - but biologically plausible - hypotheses of how fish might implement the hardware necessary for directional hearing. The combined application of pressure phase and particle motion sensing (implemented by two independent sensors and pathways) is convincing and persuasive.
- 3) They provide the ultimate test by cleverly applying the two different stimulus components with independent control. This allows for an elegant test of their hypothesis. The ultimate - and most persuasive - experiment is the selective inversion of pressure polarity by an additional opposing pair of speakers along the longitudinal axis. The application of this unnatural stimulus inverts the relative polarity between pressure and particle velocity and allows for the final test and confirmation of the hypothesis by eliciting startles in the opposite direction where fish now approach the active speaker. This is very elegant indeed!

Implementing all experimental aspect of this project is a technical and conceptual tour de force and a truly heroic effort. The manuscript is beautifully written, and the complex and intricate topic is explained clearly. The author have managed to make a potentially dry and mathematically involved matter readily accessible to a broader audience.

Overall, I find this work constitutes a significant breakthrough in a long standing question on directional hearing in teleost fish and in aquatic animals in general. It provides a final and definitive answer to a long standing and important puzzle, and it will undoubtedly have a lasting impact on the field. I firmly believe that this paper will become a 'classic', it will be of great interest to the community and I support publication strongly, provided my concerns below can be addressed.

Thank you very much for this positive appraisal, we did our best to address the remaining points.

Comments:

Major:

1) Currently the weakest aspect of the manuscript is the rejection of the P-ILD model. This is compatible with the behavioral data, but requires two separate pressure sensors – one closer to the speaker than the other. The authors' basis for rejecting this model is that they do not find two such pressure sensors in their vibrometry imaging experiments. While the imaging technology is novel and elegant, and the execution thorough and impressive, the main result still remains a negative finding. Just because you cannot find something doesn't mean it isn't there. As such the results presented in Figure 3 are certainly compatible with the hypothesis but they do not provide as conclusive a proof as the behavioral experiments, where especially the "trick" configuration is particularly elegant and convincing. I am not sure I fully and completely understand the logic of the various speaker configurations as shown in Figure 2c, but the way I see it, the vi) "Trick" configuration could be improved as follows: use the M-only version (Fig. 1c v) to deliver the particle motion cue, instead of a single speaker. , i.e. a pair of "horizontally" placed speakers instead of just a single one, as currently depicted in vi). Then a clean particle motion stimulus can be delivered with minimal pressure contamination, and the pair of "vertically" placed speakers that deliver the inverted pressure wave, would be unincumbered by the single speaker that currently provides interference of the pressure wave. This interference might also explain the somewhat poorer performance in the "trick" experiments (67% vs 80%)? Importantly, if this reasoning makes sense, then this suggested extended configuration also would solve the inversion of the pressure gradient issue; since then the pressure gradient would just be nose-to-tail and not left-to-right.

That is a very interesting suggestion, which immediately motivated us to test "trick 2.0" experimentally. We observed startles into the directions predicted by Schuijff's model:

Rebuttal Figure 1: **a**, Illustration of the sound configurations suggested by the reviewer. **b**, Centered startle trajectories pooled over all 49 tested fish in tank coordinates (without mirroring of trajectories). Left: 61% of startles to the right, two-sided binomial test: $p = 0.01$, startle trials in $N = 45$ fish. Right: 74% of startles to the left, two-sided binomial test: $p = 7 \times 10^{-9}$, startle trials in $N = 46$ fish.

It took us a while, however, to realize that the suggested configuration does not in fact predict a different response for the P-ILD model. This initially surprising realization helped us gain a better intuition and formalism for the predictions of P-ILD and P-ITD, which we believe improved the clarity of the manuscript. The new Extended Data Figure 8 summarizes this insight by illustrating the different stimulation conditions with the help of phasor plots. Since the total sound field at any point in the arena is the linear sum of the sound fields emitted from each of the four speakers, we use arrows in the complex plane to show how phase and amplitude from each speaker add up during the different stimulation conditions:

Extended Data Fig 8 | Sound monopoles and sound configurations: a, Pressure level (dashed line) falloff next to a sound monopole at several phase snapshots of a propagating wave. Left: Falloff of a 1 kHz wave over

1.5m. Right: The falloff across the width of the fish at 3 cm distance to a monopole sound source stems from the level falloff with distance. **b**, Both amplitude ratio and relative phase between pressure and particle velocity change in a distance-dependent manner. Both sound directions (-x, x) stay separate in the relative phase between pressure and particle velocity. **c-f**, Results of a simple model used to illustrate level and phase of pressure and motion along the horizontal x-axis of the inner tank. The idealized speaker activations in the different sound configurations are modeled as sinusoidal monopole sound sources (see pressure equation in **b**) located 6 cm away from the origin with frequency $f = 780$ Hz and speed of sound $c = 1500$ m/s. Acceleration is calculated from the spatial pressure gradient along the x-axis. The top rows are phasor representations of pressure or motion at five positions along the horizontal axis as indicated in the left cartoon. ILDs and ITDs are computed across a distance of $600 \mu\text{m}$ centered at the origin. See also Figure 4 for a summary on ILD and ITD across sound configurations. **c**, In the single speaker configuration, P-ILD, P-ITD, M-ILD, and M-ITD could be interpreted as rightward cues by the fish. M-ITD is even smaller than P-ITD as motion phase propagates slower than pressure phase in the near field. **d**, In the trick configuration, P-ILD and P-ITD are inverted, while M-ILD and M-ITD remain unchanged as compared to the single speaker configuration. **c-d**, See Methods section on interaural cues for comparable P-ILD and M-ILD measurements in our setup. Note that we model monopoles in open water here, but the actual speakers are extended pressure sources in a tank. **e-f**, In both the pressure configuration and the particle motion configuration P-ILD, P-ITD, M-ILD, and M-ITD are zero or undefined.

The corresponding plot for “trick 2.0” would be:

Rebuttal Figure 2: Phasor plots as well as pressure and particle motion along the left/right axis in the “trick 2.0” condition. For details compare new Extended Data Fig. 8.

By comparing the updated Extended Data Figure 8 with Rebuttal Figure 2 one can see that the new trick condition leads to similar predictions as the old one, except for the motion gradient. The new left trick condition would add the following columns in Figure 4:

Rebuttal Figure 3: Model predictions for the “trick 2.0” configurations.

Therefore, the additional experiments do not help in rejecting additional hypotheses. They were nevertheless very useful, because they motivated the new model in Extended Data Fig. 8. In addition to hopefully improving clarity, it also made us realize a mistake in our previous P-ITD derivation in that neither trick condition excludes P-ITD (see also updated Figure 4).

This brings us back to the vibrometry results and the question of how we reject both models. Both P-ILD and P-ITD models require two pressure sensors¹ (see Extended Data Figure 8). The only structures in fish that are known to pick up pressure are compressible gas-filled organs.

Given that gas-filled structures would show up on micro-CT, we are not aware of any candidate pressure sensors other than swim bladders. However, we only found one channel of pressure information being transmitted via the Weberian Apparatus to the inner ear, in line with the current understanding of otophysian sound perception.

A last option then, is that the anterior and the posterior swim bladder are used as two independent pressure sensing organs. Even though mechanical coupling to the inner ear is only known for the anterior swim bladder, it is a theoretical possibility that somehow vibration from the posterior swim bladder is separately transmitted to the brain, e.g. via an unknown sensory neuron. This would theoretically allow the fish to sense the difference in absolute pressure along the anterior-posterior axis, but not along the medio-lateral axis tested in our experiments. To further exclude that fish used a misalignment between their body axis and the North-South axis of the experimental arena, we re-analysed our data for those startles in which fish were aligned within $< 10^\circ$ or $< 3^\circ$ with the top-bottom axis of our tank. The results of this analysis are shown in Extended Data Fig. 12 (excerpt below) and are consistent with the analysis for the complete dataset.

¹ Note that pressure-gradient sensors are not sufficient for P-ILD either. P-ILD requires two pressure sensors or sensors for the pressure-*amplitude*-gradient. Sensing the difference in raw pressure (not the amplitude of its oscillation), due to the acoustic wave equation, would be equivalent to motion sensing, which we already excluded.

Summarizing these considerations and the new analysis, we added the following paragraph to the Results:

Could *Danionella c.* have another pressure sensor that the vibrometry measurements did not detect? To rule out *Danionella c.* using a pressure difference between anterior and posterior swim bladder, we repeated our behavioral analysis for only those startles in which the anterior-posterior axis of the fish was near-orthogonal to the axis of sound presentation, giving equivalent results (see Extended Data Fig. 12a). Theoretically, *Danionella c.* might possess other, unknown pressure sensors to implement P-ILD or P-ITD. However, all known sound pressure sensors in fish are based on compressible gas-filled organs. Since gas-filled structures have a high micro-CT contrast, hypothetical pressure sensing organs would either have to be microscopic to evade detection, or they would have to be based on an unknown principle of sound pressure transduction without compressible gas. Neither of these options are supported by our current knowledge of fish biology and physics of sound.

We therefore reject P-ILD and P-ITD as plausible mechanisms. Instead, the *Danionella c.* anatomy is well-suited for implementing Schuijf's model for directional hearing.

2) Success of the lateral line ablations is confirmed by an absence of staining. This is not ultimately convincing, since, again, this is a negative result. Is it possible to add an additional control that uses a functional metric? Is there a behavior that the fish can only execute with intact neuromasts? Or is it possible to record some other physiological response whose absence after ablation would provide a more convincing result?

To validate our neomycin lateral line ablation protocol, we performed DASPEI staining of hair cells, a common control to confirm lateral line ablation. A detailed study (Harris et al. J Assoc Res Otolaryngol. 2003) compared DASPEI staining to scanning electron microscopy data, hair cell selective antibody-immuno-stainings, and F-actin staining. They report that in DASPEI negative samples, they still identify ~25% of the hair cells/neuromast using other methods, in support of the reviewer's concern. Among other explanations, they note that DASPEI is a functional mitochondrial dye and would not label hair cells that have lost their mitochondrial membrane potential. Thus, some DASPEI-negative hair cells might be visible morphologically, but functionally dead. Harris et al. conclude that "DASPEI scoring provides a rapid, reliable screening method for assessment of lateral line hair cell survival."

Nevertheless, an additional functional metric could further support successful lateral line ablation. We reasoned that lateral line ablated fish might be more prone to bumping into an aquarium wall during a startle. Thus, we re-analysed our startle data and quantified wall contacts. As predicted, we saw significantly more wall-contacts after startles in the neomycin-treated group (see updated Extended Data Figure 9g). Of course, this result alone is no definite proof of complete lateral line ablation (we are not aware of any behavioral test that can fully rule out incomplete lateral line ablation), but nevertheless it provides a functional metric that has been associated (Mirjany et al. J Exp Biol. 2011) with successful lateral line ablation.

As a second functional metric, we performed a new experiment to quantify the artemia capture rate of both neomycin-treated fish and control fish in the dark. With lateral line present, they can reliably catch artemia, while the capture rate of neomycin-treated fish is heavily impaired (see updated Extended Data Figure 9h).

In addition to (a) neomycin-ablation with DASPEI control staining, (b) the wall-contact analysis and (c) Artemia-capture analysis described above, we would like to point out that (d) our stimuli were high-pass filtered above 200 Hz, whereas the lateral line is tuned to infrasound and low frequencies well below 200 Hz (references 46-49).

minor concerns/suggestions:

can the behavioral experiments shown in Figure 1 be repeated in darkness? I realize that the short delay and the symmetry in the set-up make it unlikely that the fish use some "hack" based on cryptic visual cues - but it would be a nice control.

In addition to short delay and setup symmetry, another argument against visual cues is the fact that fish escape towards the active speaker in the "trick" condition. To address the remaining possibility that there may be some unknown cryptic visual influence on the directionality of the startle response, we repeated a slightly modified version of the single speaker playback experiment from Figure 1 in the dark.

In the dark, we faced the problem that *Danionella c.* no longer swam along the y-axis, as our white wall cue that prompts the fish to oscillate along this axis was no longer effective and

fish spent more time near the walls. Consequently, the fish were less likely to encounter the trigger zone and if they did, it was less likely that they were oriented along the y-axis, a requirement for sound triggering to test left/right startle decisions.

To counteract this problem of reduced playback trigger events, we restricted fish to a central area of 4 cm x 4 cm, by placing additional transparent sheet constraining walls at the center of the inner tank. We reduced the size of the trigger zone to the center of the constraining walls (2 cm x 2 cm), so that we still left ~ 2 cm of space for the fish to perform the startle (see excerpt of newly added Extended Data Figure 12b below).

We found that lateral-line intact fish still perform directional startles in the dark (Extended Data Figure 12c). Hence, we can exclude a visual explanation for the observed directional hearing behavior.

Excerpt of Extended Data Figure 12: **b-c**, Replication of the experiment in the dark to exclude the possibility of unknown visual cues. **b**, Top view of the modified setup for playback experiment in the dark. In the dark, the white inner tank walls could no longer prompt *Danio* to swim orthogonal to the left/right axis. To nevertheless trigger playbacks at the center of the tank with the fish being aligned orthogonally within a 45° cone, we added an additional constraining tank made from thin transparent plastic, thereby increasing the likelihood of triggering playback. **c**, Centered startle trajectory for experiment in the dark. Left: activation of the left speaker leads to rightward startles (71%, two-sided binomial test: $p = 9 \times 10^{-5}$, $n = 92$ startle trials in $N = 43$ fish). Right: activation of the right speaker leads to leftward startles (69%, two-sided binomial test: $p = 0.0004$, $n = 93$ startle trials in $N = 43$ fish)

The numbering of the Extended Data Figures do not appear in ranked order in the text.

Indeed, thank you. We checked and fixed the order.

Referee #3 (Remarks to the Author):

Veith et al provide a technical tour de force and comprehensive paper on sound localization in a tiny transparent species that allows visualization of the brain and otolith organs within an intact fish. The fish bioacoustics world is divided into workers with specializations in engineering, acoustics, auditory and muscle physiology, experimental anatomy and histology, behavior and ecology. Rarely are more than two of these fields represented in a single paper. The authors use expertise in all of these fields (save ecology) to attack a difficult question, namely how a fish localizes sound, which has been a major unsettled issue in fish acoustics. The question is complicated due to the high speed of sound in water, which reduces cues that would be available to terrestrial animals and humans. Among other accomplishments, the authors are able set up different test conditions to differentiate between previous hypotheses. They provide support for Schuijf's notion that otophysans (fishes with Weberian ossicles that connect the swimbladder to the ears) utilize a combination of particle velocity and acoustic pressure. If memory serves, Schuijf's theory, previously hypothetical, has now been supported by the study whereas other competing ideas failed experimentally. The authors also accomplish this work in fish with intact and disabled lateral lines. Lateral lines potentially provide directional auditory cues although they are primarily tuned to lower frequency water movement. In this case the ears are sufficient and the lateral line is not required for a directional escape response. Among other feats, the authors use laser vibrometry to demonstrate differences in how the otoliths respond individually to pressure or particle velocity. Such work would previously have required anesthetic and surgical exposure that could affect vibratory behavior of end organs.

The paper is challenging due to the inclusion of a large amount of diverse information but is extremely well written, clear and does a good job of summarizing past literature.

I have a couple of minor suggestions.

Introduction, para 3. Delete plainfin or put it before midshipman.

Thank you for catching this, it has been corrected (==> plainfin midshipman)

Fig. 2. The authors provide an illustration of the stimulus waveforms. Given the acoustic complexity of small underwater tanks, I suspect the actual signals were not quite this pretty. I think a statement might be warranted. I also note that the results stand so the pressure and velocity signal were clearly sufficient to support the results in the paper.

We added a reference to the actual traces (shown in Extended Data Figures 2 and 3 (previously 9 and 10)) to the caption of Figure 2.

Behavioral set up and protocol. Pg 16.

I was originally going to ask for a little more detail such as dimensions of the outer tank, but I note that much of this is given in the figure. I think it would be worthwhile to specify the distance between the speaker and the choice point, how the speaker was mounted, and the effect of the plastic inner tank on acoustic transmission.

The mounting of the speakers is an important piece of information that is now added to the methods. We also specified the distance of the speakers to the inner tank wall. The actual distance to the fish upon playback is then variable, and depends on the position of the fish inside the trigger zone. However, our sound calibration always fixed the pressure and motion phase and amplitude ratio to that of a sound monopole at 3 cm distance (described in the methods section on sound stimulation waveforms). We address the effect of the inner tank on acoustic transmission below.

The plastic inner tank likely has an acoustic impedance similar to water and would not be a major problem. Still, it would be nice to state how signal amplitude and waveform were or were not affected.

We agree that the polypropylene sheet wall is roughly acoustic-impedance matched to water. Additionally, the sheet is very thin (160 μm), an information we now added to the methods section. Finally, our calibrated sound conditioning method which accurately delivers target sounds to the fish position would have compensated effects introduced by the plastic sheet.

As our setup is built such that the sheet walls are just clamped and easily removable, we could readily test the reviewer's concern that the plastic sheet may affect the sound. To this end, we measured the impulse response of a speaker in presence and absence of the sheet in-between speaker and hydrophone in the setup used for the experiments. We did not detect a difference in impulse response between the two scenarios (see figure below).

Rebuttal Figure 5: Impulse response with and without the inner tank wall made of 160 μm thick polypropylene sheets. Hydrophone recordings of 200 playback trials in each condition show no difference in impulse response in presence or absence of the plastic sheet wall.

What does transparent ground of the inner tank mean? Fish column is not exactly clear.

We updated this section of the Methods and clarified that the inner tank was made out of thin polypropylene sheets. The bottom of the inner tank was made out of a transparent sheet (we replaced “ground” with “bottom”).

The inner tank had a body of water, a water column, that contained the fish. We replaced our “fish column” neologism with the more appropriate “water column” and clarified its meaning.

Reviewer Reports on the First Revision:

Referees' comments:

Referee #1 (Remarks to the Author):

The authors' revisions are excellent. I won't repeat my comments from the original review, except to note that this is a landmark study, taking advantage of a new fish model and new laser scanning techniques to resolve long standing questions about how fish hear.

Referee #2 (Remarks to the Author):

looks really good!

Referee #3 (Remarks to the Author):

I think the authors have done an excellent job of responding to the criticisms and have written a landmark paper answering formerly intractable questions.

In terms of your query about statistics, I would say the following. The authors are working in a difficult area, namely
can a fish localize a sound in a small tank. The major purpose of the paper is to demonstrate the mechanism of localization in a fish with lateral lines, pressure reception courtesy of the swimbladder and linkage to the ears via Weberian ossicles, and finally direct stimulation of the ears caused by particle velocity. They succeed handily in this purpose. Further work could refine the precision of localization (perhaps using circular statistics common in studies on orientation to the sun, magnetic fields, etc), but that would be another paper quantifying precision and less likely to interest Nature. I would say that a sentence or two on their use of the binomial test (mentioned in the results) could be added to the methods, but that is a minor addition. The binomial test is sufficient to support their results.

Michael L Fine

Author Rebuttals to First Revision:

Referee #1 (Remarks to the Author – 2nd round):

The authors' revisions are excellent. I won't repeat my comments from the original review, except to note that this is a landmark study, taking advantage of a new fish model and new laser scanning techniques to resolve long standing questions about how fish hear.

Referee #2 (Remarks to the Author – 2nd round):

looks really good!

Referee #3 (Remarks to the Author – 2nd round):

I think the authors have done an excellent job of responding to the criticisms and have written a landmark paper answering formerly intractable questions.

In terms of your query about statistics, I would say the following. The authors are working in a difficult area, namely can a fish localize a sound in a small tank. The major purpose of the paper is to demonstrate the mechanism of localization in a fish with lateral lines, pressure reception courtesy of the swimbladder and linkage to the ears via Weberian ossicles, and finally direct stimulation of the ears caused by particle velocity. They succeed handily in this purpose. Further work could refine the precision of localization (perhaps using circular statistics common in studies on orientation to the sun, magnetic fields, etc), but that would be another paper quantifying precision and less likely to interest Nature. I would say that a sentence or two on their use of the binomial test (mentioned in the results) could be added to the methods, but that is a minor addition. The binomial test is sufficient to support their results.

Thank you for this suggestion. We added the following sentence to the Methods:

“Using the two-sided binomial test, we calculated how likely a measured directional bias (approach or escape) would have been observed if the response was unbiased.”